# Molecular traces of *Drosophila* hemocytes reveal transcriptomic conservation with vertebrate myeloid cells

Sang-Ho Yoon[1,2,3], Bumsik Cho[1], Daewon Lee[1], Hanji Kim[1], Jiwon Shim[1,2,3,4]*, Jin-Wu Nam[1,2,3,4]*

1 Department of Life Science, College of Natural Sciences, Hanyang University, Seoul, Republic of Korea, 2 Hanyang Institute of Advanced BioConvergence, Hanyang University, Seoul, Republic of Korea, 3 Hanyang Institute of Bioscience and Biotechnology, Bio-BigData Research Center, Hanyang University, Seoul, Republic of Korea, 4 Research Institute for Convergence of Basic Sciences, Hanyang University, Seoul, Republic of Korea

* jshim@hanyang.ac.kr (JS); jwnam@hanyang.ac.kr (J-WN)

**Data Availability Statement:** The single-cell dataset generated in this study has been deposited in the NCBI Gene Expression Omnibus (GEO) repository under the accession number

## Abstract

*Drosophila* hemocytes serve as the primary defense system against harmful threats, allowing the animals to thrive. Hemocytes are often compared to vertebrate innate immune system cells due to the observed functional similarities between the two. However, the similarities have primarily been established based on a limited number of genes and their functional homologies. Thus, a systematic analysis using transcriptomic data could offer novel insights into *Drosophila* hemocyte function and provide new perspectives on the evolution of the immune system. Here, we performed cross-species comparative analyses using single-cell RNA sequencing data from *Drosophila* and vertebrate immune cells. We found several conserved markers for the cluster of differentiation (CD) genes in *Drosophila* hemocytes and validated the role of *CG8501* (*CD59*) in phagocytosis by plasmatocytes, which function much like macrophages in vertebrates. By comparing whole transcriptome profiles in both supervised and unsupervised analyses, we showed that *Drosophila* hemocytes are largely homologous to vertebrate myeloid cells, especially plasmatocytes to monocytes/macrophages and prohemocyte 1 (PH1) to hematopoietic stem cells. Furthermore, a small subset of prohemocytes with hematopoietic potential displayed homology with hematopoietic progenitor populations in vertebrates. Overall, our results provide a deeper understanding of molecular conservation in the *Drosophila* immune system.

## Author summary

The immune system protects organisms from invaders and has been conserved throughout animal evolution. Hemocytes are blood cells in *Drosophila* that are known to have similar functions to human innate immune cells, but the relationship between *Drosophila* and other species has only been predicted with a few genes. Here, we integrate large public *Drosophila* larval hemocyte datasets to define rare cells, consensus cell types, and states. We then perform a comprehensive comparative analysis of *Drosophila* hemocytes with

GSE184781 (https://www.ncbi.nlm.nih.gov/geo/query/acc.cgi?acc=GSE184781).

**Funding:** This work was supported by the National Research Foundation (NRF) of Korea, which is funded by the Ministry of Science & ICT (2020R1A4A1018398, 2021R1A2C3005835, 2022M3E5F1018502, and RS-2023-00207840) to J.-W.N., (2019R1A2C2006848 and RS-2023-00218602) to J.S., and (2020R1A6A3A13076391) to S.-H.Y. The funders had no role in study design, data collection and analysis, decision to publish, or preparation of the manuscript.

**Competing interests:** The authors have declared that no competing interests exist.

immune cells from zebrafish, mice, and humans, revealing that a phagocytic cell type in *Drosophila*, plasmatocytes, is conserved as a myeloid cell in other organisms. We also report that a novel plasmatocyte marker gene, *CG8501*, which is conserved as in human as CD59, functions in the formation of normal NimC1+ plasmatocytes for bacterial uptake. Our work provides the first transcriptome-wide analysis between *Drosophila* and vertebrate species and documents the conservation of orthologous genes and cell types in *Drosophila* hemocytes.

## Introduction

The immune system, consisting of innate and adaptive immunity, has evolved to protect organisms from the various pathogens they may encounter throughout their lives. Innate immunity, the older system, can be traced back to invertebrates, which split from vertebrates more than 500 million years ago [1]. *Drosophila* is one of the most extensively studied model organisms, and its blood cells, also known as hemocytes, are often considered myeloid-like cells that play several roles in the innate immune system, including the phagocytosis of pathogens [2] and tissue remodeling [3,4].

Fully differentiated *Drosophila* hemocytes have been classified into three morphologically distinct populations with different functions: plasmatocytes (PMs), crystal cells (CCs), and lamellocytes (LMs). The most abundant cell type of hemocytes is the plasmatocytes, which are described as macrophage-like cells due to their phagocytic functions [5,6], while CCs, a minor population characterized by crystalline inclusions in the cytoplasm, induce melanization during the wound healing process [7]. The LMs are a specialized cell type that is differentiated in reaction to parasitic infection, such as wasp infection [8]. In processes similar to those described in vertebrates, *Drosophila* hemocyte development occurs through two different hematopoietic waves: embryonic hematopoiesis, in which hemocytes originate from the head mesoderm and circulate during larval development, and lymph gland hematopoiesis, in which hemocytes arise from the larval cardiac mesoderm and eventually dissociate into circulation during pupariation [9,10]. Differing from hemocytes originating from the embryonic hematopoiesis, the lymph gland houses hemocyte progenitors, called prohemocytes (PHs), that give rise to mature hemocytes and are maintained by the microenvironment niche cells of the posterior signaling center (PSC) [11]. In addition to the known hemocyte types, GST-rich cells and adipohemocytes have also been characterized in the lymph gland owing to the development of single-cell transcriptome analysis [12]. The PHs represent a heterogeneous population of progenitor cells depending on the degree of differentiation; a very small fraction of cells defined as PH1 (stem cell-like) differentiate into all of the above cell types. *Drosophila* hematopoiesis has been suggested as a valuable model system for studying immune responses to diseases [13]. However, the relationship between *Drosophila* hemocytes and those in vertebrates has heavily relied on functional homologies described by a handful of marker genes, and a systematic analysis at the transcriptome level has yet to be undertaken.

In this study, we analyzed 43,891 *Drosophila* hemocytes originating from lymph glands or in the circulation system in wild-type and wasp-infected larvae using single-cell RNA sequencing (scRNA-seq) in conjunction with publicly available zebrafish, mouse, and human scRNA-seq data ($n$ = 281,099 cells) to investigate cross-species cell type similarities. We first compared *Drosophila* genes with cluster of differentiation (CD) markers and identified conserved sequences between *Drosophila CG8501* and *CD59* in vertebrates. Loss of *CG8501* expression was associated with a decrease in Hml$^+$ hemocytes and aberrant bacterial uptake. In a

transcriptome-wide comparative analysis, we revealed conservation between *Drosophila* hemocytes and vertebrate innate immune cells, especially macrophages. *Drosophila* PH1 cells were homologous to progenitor populations in vertebrates, supporting the multipotent progenitor role of this cell type. Our work provides the first transcriptome-wide view of similarities between *Drosophila* hemocytes and vertebrate immune cells.

## Results

### Integration of *Drosophila* hemocyte scRNA-seq data

In our previous studies [12,14], we sequenced the transcriptomes of *Drosophila* lymph gland and circulating hemocytes at various timepoints during development using the droplet-based single-cell sequencing platform, Drop-seq [15], and identified diverse subclusters and developmental trajectories. However, hemocyte populations in the lymph gland or circulating hemocytes alone do not represent the entire hemocyte population in *Drosophila* larvae. To build a comprehensive hemocyte single-cell atlas of larval hemocytes, we integrated whole transcriptomes of hemocytes both in circulation and in the lymph gland 72, 96, and 120 hours after egg laying (AEL). Additionally, we combined lymph gland hemocyte data from wasp-infected larvae at 96 h AEL, 24h post-infection, and circulating hemocyte data at 96 and 120 h AEL, 24 h and 48 h post-infection, respectively, in which lamellocyte populations are largely visible as the immune system is triggered (Figs 1A and 1B and S1A). A total of 43,933 cells from seven major cell types were collected, with median counts of 5740 unique molecular identifiers (UMIs) and 1467 genes per cell (S1B and S1C Fig). Briefly, 33.47% (*n* = 14,705) and 49.90% (*n* = 21,923) of cells were annotated as prohemocytes and plasmatocytes, respectively. Because PH1 has been previously reported to possess stem cell-like functions, and because they independently clustered in our analysis (Fig 1A) [12], we separated prohemocyte subcluster PH1 from the rest of the prohemocytes in both lymph glands and circulation transcriptomes. Additionally, we distinguished plasmatocytes specific to the lymph gland at the 120 h AEL time point. These 120 h AEL-specific plasmatocytes expressed additional non-classical plasmatocyte markers, such as *CG8501* or *Ama* (Fig 1D and 1E). Large compositional differences between the lymph gland and circulation transcriptomes were identified; the majority of prohemocytes were found in the lymph gland (12,357 out of 14,705 cells), and all adipohemocyte cells were exclusively annotated in the lymph gland (Fig 1C). A small number of PSC-like cells were also found in circulating hemocytes (Fig 1C, 42 out of 313 cells). Lamellocytes, the third largest population in our dataset (12.36%), predominantly originated from wasp-infected larvae, especially from circulating hemocytes obtained at 120 h AEL (48 h post-infection; 4355 out of 5431 cells). This observation indicates the specialized defensive role of this cell type during parasitic wasp infections (Fig 1C and 1D). Interestingly, GST-rich cells were also found in circulation, but exclusively in datasets obtained during wasp infection. Given that the number of GST-rich cells was also increased in a wasp-infected lymph gland counterpart in our previous report [12], this suggests a potential association between GST-rich cells and lamellocyte differentiation. Again corroborating our previous research [12], several lineage-specific markers were identified, and the top-expressing marker genes were shared between cell types from the lymph gland and in circulation (Fig 1E and S1 Table). We also explored lists of the curated marker genes of major hemocytes that are largely expressed by corresponding cell types (S2 Fig) [16]. Next, we expanded our analysis to incorporate publicly available scRNA-seq data that encompass lymph glands or circulating hemocytes in *Drosophila melanogaster* (S3A Fig and S2 Table) [14, 17–20]. Initially, we annotated cells based on the annotations provided in the original research papers, using label transfer. Then, we compared these annotated cells across different scRNA-seq studies (S4 Fig). Crystal cells and lamellocytes showed remarkable

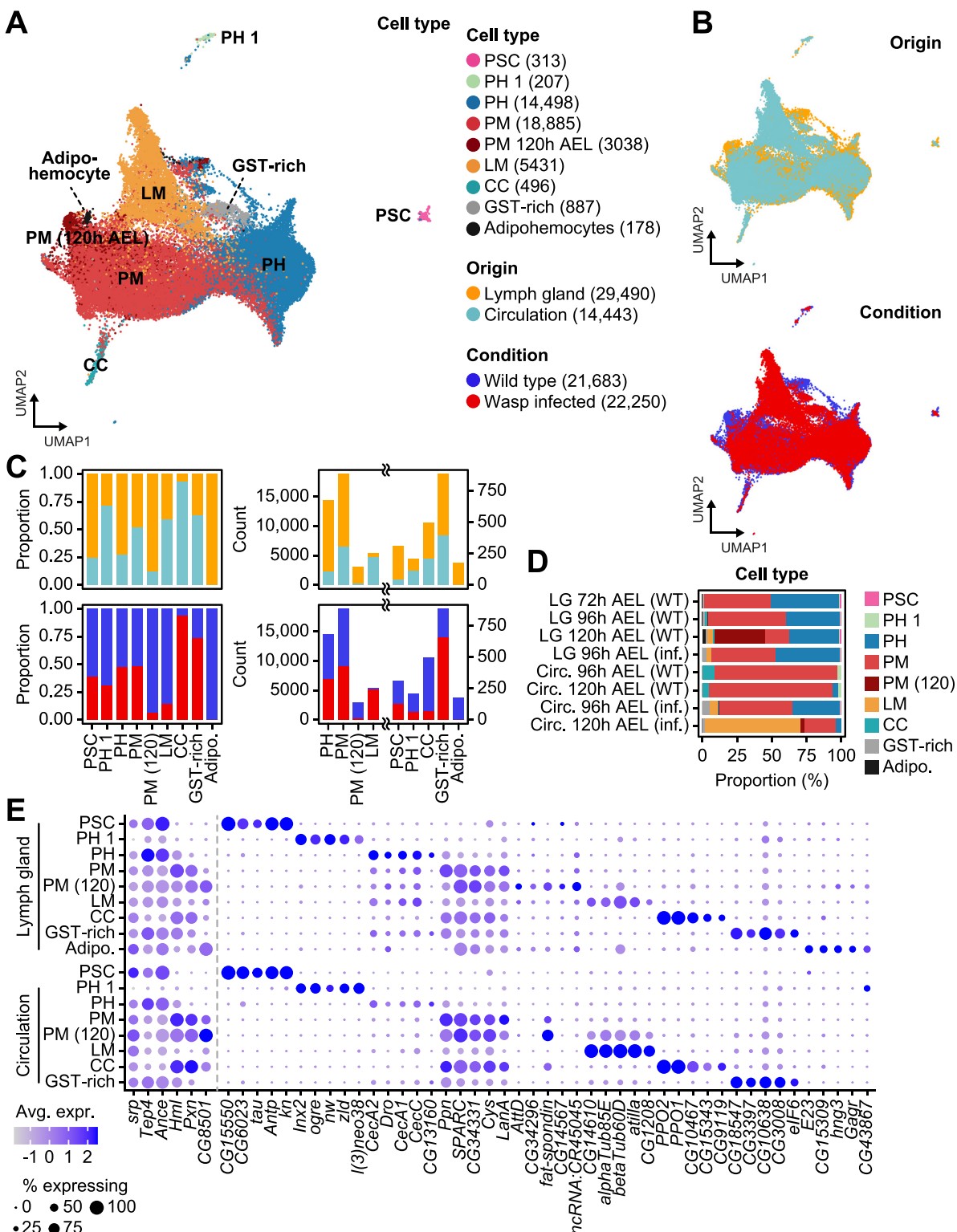

**Fig 1. Integration of *Drosophila* larval hemocyte Drop-seq datasets.** (**A**) A UMAP plot of the nine major hemocyte types identified in *Drosophila*. The cell count for each cell type is indicated in parentheses. (**B**) UMAP plots showing the tissue origins (top) and experimental conditions (bottom) of hemocytes. (**C**) The proportion (left) or count (right) of tissue origins (top) and experimental conditions (bottom) of hemocytes for each cell type. (**D**) The proportion of cell types for each sampling time point and condition, wild type (WT) or wasp-infected (inf.). (**E**) A dot plot presenting the expression of the top 5 cell type markers in the lymph gland (top) and circulation (bottom). The dot color indicates the average level of expression, and the dot size represents the percentage of cells expressing the gene in each cell type.

consistency, whereas plasmatocytes showed less agreement among different studies. This inconsistency can be attributed to differences in the criteria used for clustering analysis in different studies.

To address this issue, we combined data from six scRNA-seq studies and subjected a total of 125,402 cells to a uniform analytic pipeline. Consequently, we clustered these cells into 17 distinct cell types and transcriptional states (Figs 2A and S3B), labeling the clusters based on the expression of known markers or their top-expressing genes (Fig 2A and 2C). These 17 clusters included PSC cells, two prohemocytes (PH1 and PH), seven plasmatocytes (PM-Hml, PM-prolif, PM-AMP, PM-Gst, PM-late1, PM-late2, and PM-Lst), two lamellocytes (LM1 and LM2), crystal cells (CC), and four other types (Hsp, Unknown, Muscle, and S-Lap). Two small clusters, primarily originating from InDrops (Hsp and unknown); muscle cells; and S-Lap were excluded from the following analysis due to a lack of plasmatocyte markers or bias towards a particular dataset (Fig 2A and 2B). As seen in previous studies [14], plasmatocytes showed the highest heterogeneity. The PM-Hml cluster contained the majority of plasmatocytes (37,853 cells), consistently enriched in plasmatocyte marker expression, including *Hml*, *Pxn*, *vkg*, *Col4a1*, and *Ppn* (Fig 2C and 2D). In addition, PM-prolif was the second largest plasmatocyte cluster (14,210 cells) (Fig 2C and 2D), suggesting a highly proliferative nature of plasmatocytes. The PM-Gst cluster exhibited enrichment in glutathione S transferases, such as *GstE6* or *GstE1*, and included GST-rich cells identified in our previous study (Fig 2C and 2D). The PM-AMP cluster displayed the expression of various antimicrobial peptides—such as *Drs*, *AttB*, or *Dro*—as has been reported in other studies [17]. Notably, while the majority of PSC cells originated from lymph gland datasets (Fig 2B) [12,19], a small number of PSC cells were also found in circulating hemocytes expressing similar marker genes (Fig 2C, *CG15550*, *mthl7*, and *Antp*). The role of these PSC-like cells, also known as primocytes, requires further investigation [16,18]. Lastly, prohemocytes were initially defined as precursors of plasmatocytes in the medullary zone and formed a developmental continuum in lymph glands [12]. Some prohemocytes were also found in the circulating hemocyte populations; however, most of these cells were defined under wasp infection, implying they are either prohemocytes originated from dissociated lymph glands or plastic plasmatocytes able to de-differentiate to form lamellocytes (Figs 2D and S3C).

In summary, we constructed a comprehensive landscape of *Drosophila* hemocytes by integrating two developmental lineages from diverse time points and conditions. All clustering results and expression levels of marker genes from various conditions are available at Fly scRNA-seq Database 2.0 (http://big.hanyang.ac.kr/flyscrna).

## Hematopoietic cells in zebrafish, mice, and humans

The scRNA-seq data from hematopoietic stem and immune cells from zebrafish, mice, and humans were collected from previous studies and public single-cell atlas databases [21–25]. Specifically, data from 3301 zebrafish kidney marrow cells, obtained using the InDrop-seq platform; 6977 cells from the Mouse Cell Atlas (MCA, Microwell-seq); 8191 cells from Tabula Muris (10X Chromium, 3427; Smart-Seq2, 4764 cells); 20,158 cells from the Human Cell Landscape (HCL, Microwell-seq); and 242,662 cells from the Human Cell Atlas (HCA, 10X Chromium) were used. All datasets were newly clustered or re-clustered by species with cell annotations based on the expression levels of known marker genes from the literature and the atlas databases to facilitate comparisons (S5A–S5C Fig). To examine similarities between datasets from the same species, we transformed single-cell expressions into pseudo-bulk expressions and measured Spearman correlations by cell type (S5D and S5E Fig). Analysis of three independent mouse datasets, obtained with different sequencing platforms, showed that data

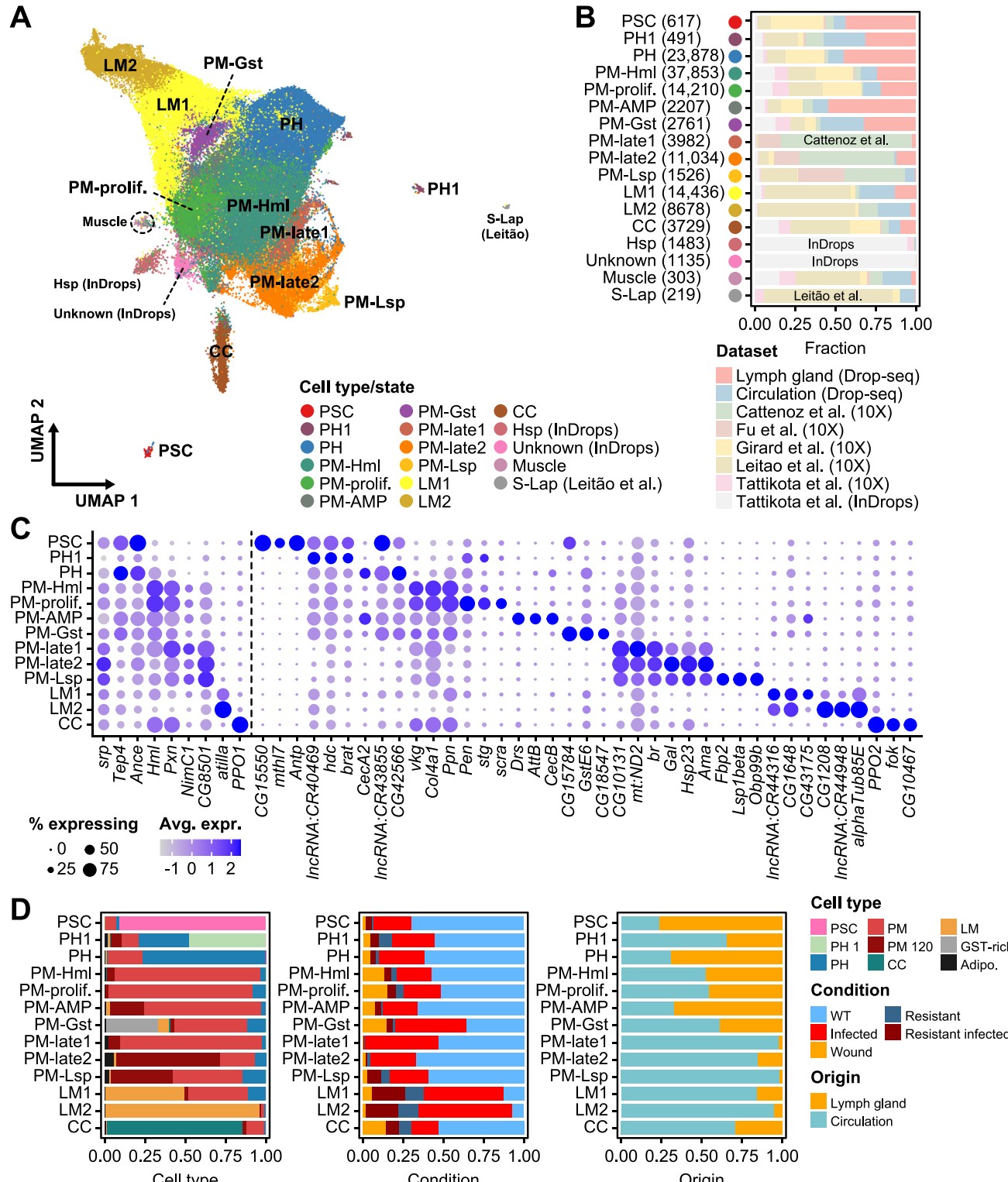

**Fig 2. Characteristics of public *Drosophila* larval hemocyte scRNA-seq datasets.** (**A**) A UMAP plot of hemocyte clusters newly identified in the integrated *Drosophila* scRNA-seq dataset. (**B**) The proportion of each cell cluster represented by each dataset. The cell count for each cell type is indicated in parentheses. (**C**) Dot plots presenting the expression of the top three markers for each cell type. The dot color indicates the average level of expression, and the dot size represents the percentage of cells expressing the gene in each cell type. (**D**) Proportions of broad cell types for each cell type/state defined in the integrative analysis (left) and categorized by experimental condition (middle) and tissue origin (right).

from mouse immune cells were well-matched across cell types, except for macrophages, which were exclusively found in the MCA (S5D Fig). Similarly, results from human immune cell types also agreed well between the two datasets, except for those from B cell progenitors and platelets, which were exclusively found in HCA (S5E Fig). The independent datasets were subsequently integrated by species, summarizing results from 13 and 16 different mouse and human cell types, with 15,168 and 262,630 cells, respectively (S5F and S5G Fig).

## Orthologous genes are sufficient to distinguish known immune cell types

To investigate transcriptomic similarities between immune cells from the four species, we removed non-hemocytes and non-immune cells and identified orthologous genes among species (Fig 3A). Orthologous genes from all *Drosophila*, zebrafish, mouse, and human pairs were extracted using the DRSC Integrative Ortholog Prediction Tool (DIOPT) database, which provides thorough reports from multiple databases with weighted scores for gene pairs (see the Methods section) [26]. This showed that 5739 genes were conserved between *Drosophila* and zebrafish and expressed in both datasets, whereas 5192 and 6474 genes were matched between and expressed in *Drosophila* and mouse and *Drosophila* and human, respectively (Fig 3B and S3 Table). The number of conserved genes continuously increased between zebrafish and mice (8714 genes) and mice and humans (10,379 genes), which is partly a result of the huge evolutionary gap between invertebrates and vertebrates and partly a result of there being fewer annotated genes in the *Drosophila* genome (17,714 genes, based on the Berkeley *Drosophila* Genome Project [BDGP] release 6.22, compared to 62,492 genes in the human genome, based on GENCODE v34). Approximately 24.09% of *Drosophila* genes (4267 of 17,714 genes) were conserved and expressed in all four species (S4 Table).

Based on this list of conserved genes, we sought to assess whether the expression levels of orthologous genes would be sufficient to distinguish different cell types in the four species. To this end, we iteratively performed a *t*-distributed stochastic neighbor embedding (*t*-SNE) [27] dimensionality reduction analysis of full or downsampled datasets using the corresponding orthologous genes (Figs 3C–3E and S6). Major cell types were well separated and clustered for all four species, indicating that orthologous gene expression levels are sufficient to characterize immune cell types. In *Drosophila*, all hemocyte types were clustered except for GST-rich cells, which were largely scattered across the prohemocyte and plasmatocyte clusters in the *t*-SNE plot obtained using the 5676 genes conserved between *Drosophila* and zebrafish (Figs 3C and S6A), suggesting that GST-rich cells might be defined on the basis of *Drosophila*-specific genes. Adipohemocytes tended to cluster outside of well-defined prohemocyte and plasmatocyte clusters; however, a few prohemocytes or plasmatocytes were grouped together. Interestingly, B and NK/T cells in zebrafish were separated from other cell types but comingled with each other (Figs 3C and S6B). These cell types were well clustered when the analysis was reperformed with the 8714 genes conserved between fish and mice (Figs 3D and S6C), suggesting that genes that perform specialized functions in adaptive immune systems exhibit conserved expression in vertebrates but are absent in *Drosophila* hemocytes. Indeed, many genes conserved only between vertebrates were enriched with biological processes related to functions or differentiation of lymphocytes and regulations of interleukin production (Fig 3F and 3G and S5 Table). We also performed a *t*-SNE analysis using orthologous genes that are shared between all four species (4267 genes) and found that the major cell types separated to a lesser extent (S7 Fig). The GST-rich cells in *Drosophila* again failed to separately cluster, and T/B cells in vertebrates either comingled or formed a continuous cluster. In summary, the expression levels of orthologous genes are sufficient to describe different immune cell types,

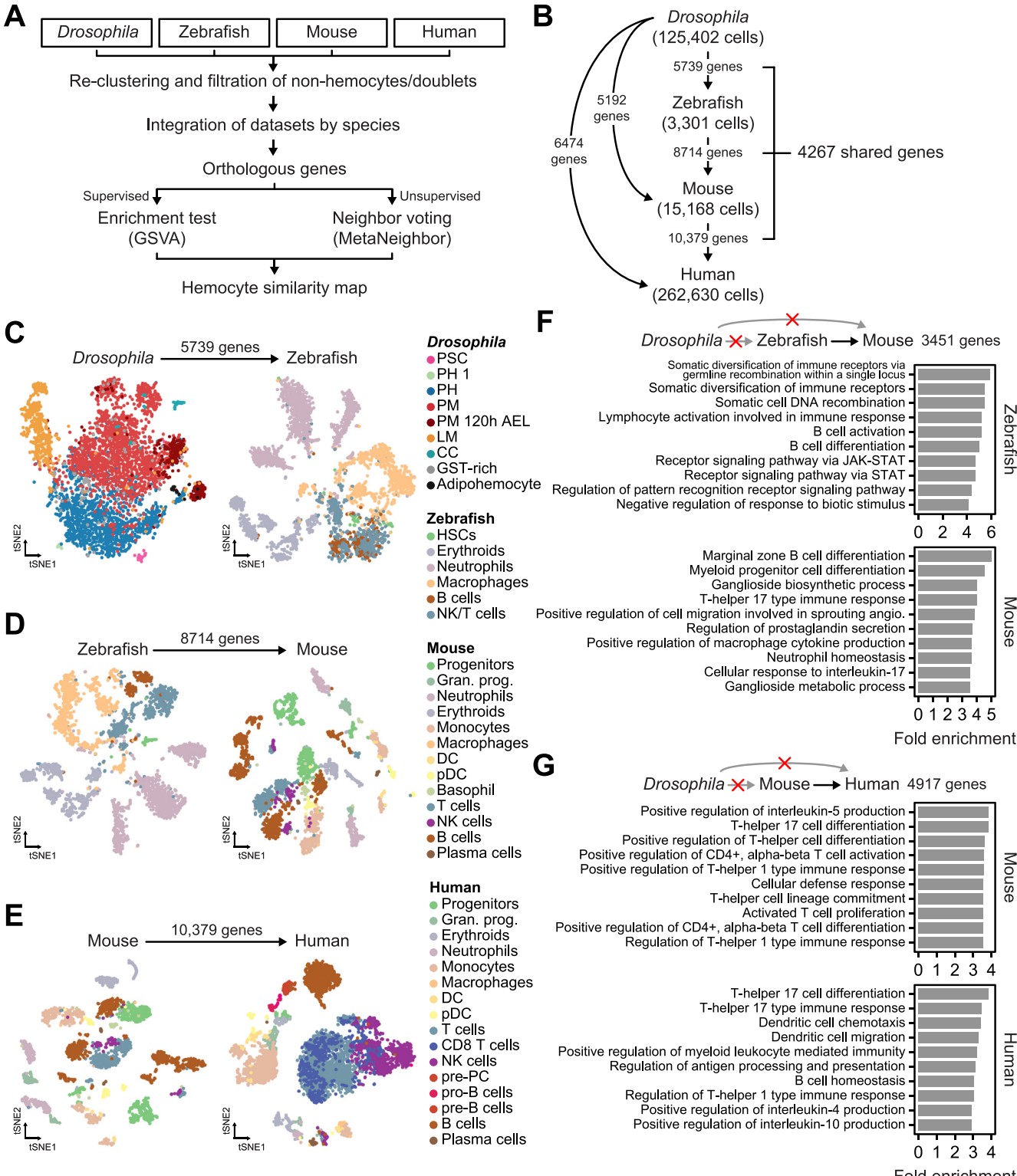

**Fig 3. Identification of cell type clusters using orthologous genes.** (**A**) The workflow of the analysis comparing immune cell types between species. (**B**) A summary of the expressed orthologous genes between each species used in this study. (**C**) The *t*-SNE plots of *Drosophila* hemocytes and zebrafish immune cells using 5739 orthologous genes. The *Drosophila* data were downsampled to one-tenth (4389 cells). Data from all 3301 zebrafish cells were used. (**D**) The *t*-SNE plots of zebrafish and mouse immune cells using 8714 orthologous genes. The mouse data were randomly downsampled to one-fifth (3034 cells). (**E**) The *t*-SNE plots of mouse and human immune cells using 10,379 orthologous genes. The human data were randomly downsampled to one-fiftieth (5253

cells). (**F** and **G**) Bar plots showing the fold enrichment of biological processes as identified by gene ontology. Only genes conserved between zebrafish and mice (**F**) or mice and humans (**G**) were tested.

but genes characterizing adaptive immune cells are largely unexpressed in *Drosophila* hemocytes.

## Cross-species comparative analyses of immune cell types

In vertebrates, the conservation of immune cells between species is well-known [28], and recent studies have shown conserved marker genes and regulatory programs between *Drosophila* and vertebrates at the whole organism level [29,30]. Although the homology between *Drosophila* hemocytes and vertebrate immune cells has been previously discussed [31], comparative transcriptomic analyses across immune cell types have not been performed. To address this issue, we compared the expression and conservation of cluster of differentiation (CD) genes and validated functions in *Drosophila* hemocytes (Fig 4). Next, we performed a supervised analysis using conserved marker genes and an unsupervised analysis using information from neighboring cells (Fig 5). The detailed analyses and experimental validations are described in the following sections.

### *Drosophila CG8501*, orthologous gene of human *CD59*

CD molecules are leukocyte markers, which play important roles in immune development and activation and are commonly used in immunophenotyping for diagnostic purposes and cell annotations [32]. In *Drosophila*, the expression of CD orthologs highlights the functional conservation of CD genes in hemocyte immunity. For example, *croquemort* (*crq*), a well-known ortholog of *CD36*, functions in the removal of apoptotic cells [2]. We searched for the conservation of CD gene markers in *Drosophila* hemocytes and found six conserved hemocyte genes, including *crq* (Fig 4A). *visgun* (*vsg*), which is widely expressed in *Drosophila* hemocytes, was indicated as a *CD164* orthologue and has been recently established as a crucial marker for phagocytosis and immune activation upon *Photorhabdus luminescens* bacterial infection [33]. The tetraspanin 42E family genes, including *Tsp42Ed* and *Tsp42Ee*, were conserved as CD63 and expressed in plasmatocytes and adipohemocytes at 120 h AEL. Another tetraspanin family gene, *Tsp96F*, showed multiple homologies with human *CD9*, *CD81*, and *CD82*. The gene *CG8501* was homologous to human *CD59* and was enriched in plasmatocytes (120 h AEL) and adipohemocyte (Fig 4A and 4B).

To test the expression of CD proteins in hemocytes, we used antibodies targeting homologous CD protein epitopes in *Drosophila*. Three antibodies against human CD proteins, including CD63, CD164, and CD59, were found to recognize *Drosophila* hemocyte proteins. While the staining for CD63 and CD164 was weak in *Drosophila* hemocytes, the expression of anti-CD59, which potentially targets CG8501, was clearly visible in the cytoplasm of circulating hemocytes (S8A Fig). Compared to the wild type (*Oregon R*), the expression of CD59 was significantly reduced when *CG8501* was inhibited either through one-copy loss of *CG8501* (*Df (2R)BSC859/SM6a*) (S8A Fig) or *CG8501* RNAi in *Hml*⁺ hemocytes (*HmlΔ-Gal4 UAS-GFP UAS-CG8501 RNAi*) (Fig 4C). Conversely, no reduction was observed in deficiency mutants containing *Tsp42E* family genes (*Df(2R)BSC262/CyO*) or *vsg* (*Df(3L)BSC393/TM6C*), which target anti-CD63 and anti-CD164, respectively (S8A Fig). These results suggest that anti-CD59 specifically recognizes CG8501 in *Drosophila* hemocyte.

To better understand the function of CG8501 in hemocytes, we investigated whether *CG8501* RNAi modified the differentiation or proliferation of embryonically derived

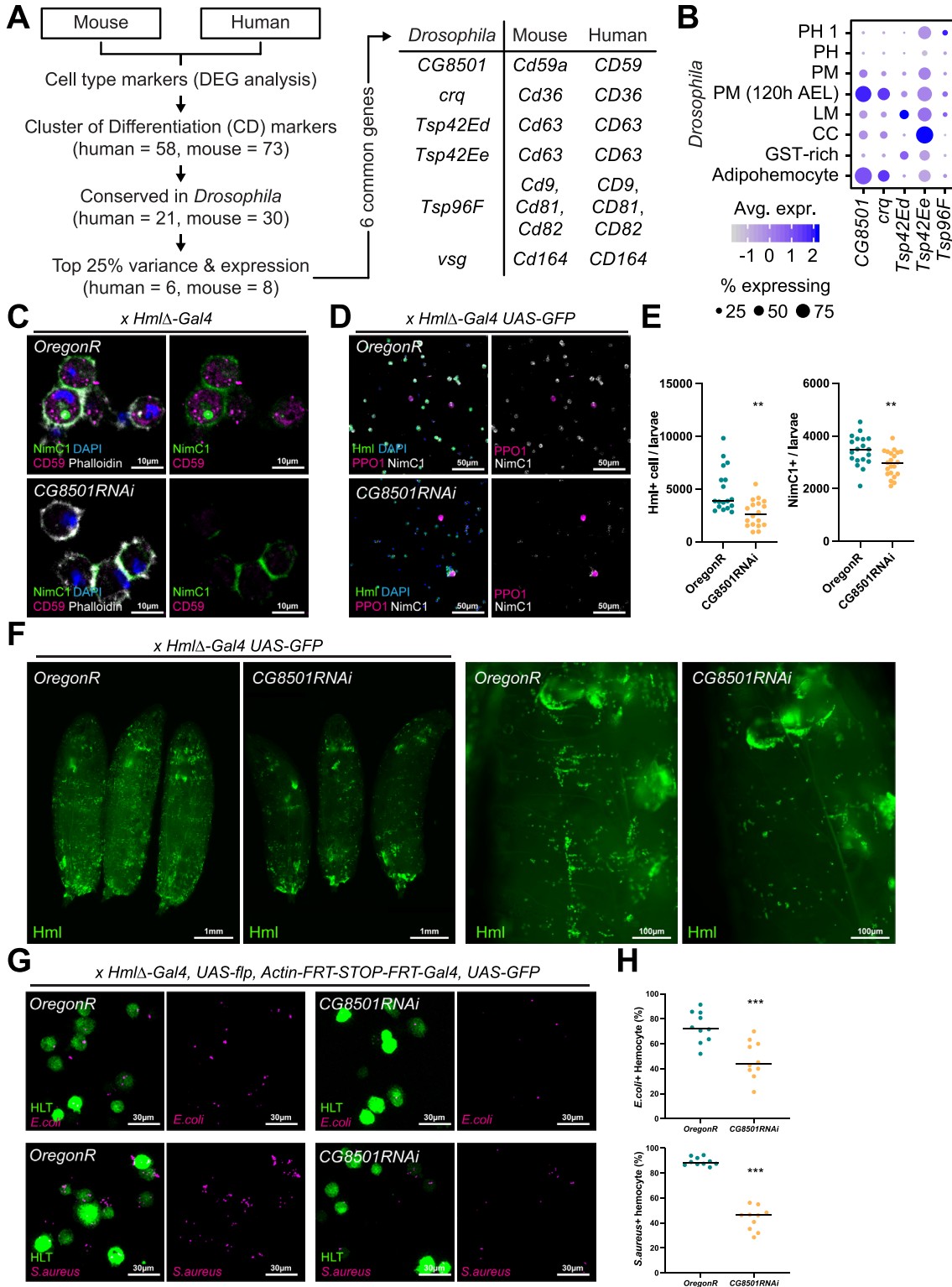

**Fig 4. *Drosophila CG8501*, an orthologous gene of human *CD59*.** (**A**) Schematic illustration of the orthologous gene selection process. (**B**) Expression of CD orthologs in *Drosophila* hemocyte sub-populations. The dot color indicates the average level of expression, and the dot size represents the percentage of cells expressing the gene in each cell type. (**C**) Expression of protein CG8501 in the hemocyte detected by antibody staining against human CD59 protein. Protein CG8501 (magenta) was expressed in the cytosol and did not overlap with NimC1 (green) or phalloidin (white). Nuclei were stained by DAPI (blue). (**D**) Decrease in Hml+ hemocyte numbers in *CG8501*

RNAi expressing mutants. Compared to wild-type hemocytes (*HmlΔ-Gal4 UAS-GFP Oregon R*), knockdown *CG8501* hemocytes (*HmlΔ-Gal4 UAS-GFP CG8501 RNAi*) show low *Hml* (green) and NimC1 (white) expressions. However, PPO1 (magenta)-positive mature crystal cells or the number of total hemocytes (DAPI, blue) did not change. (**E**) Quantification of Hml+ or NimC1+ hemocyte numbers in wild-type hemocytes (*Oregon R*) and knockdown *CG8501* hemocytes (*CG8501RNAi*) (**$p < 0.001$). Horizontal bars indicate median values. (**F**) Whole mount images of wild-type larvae (*Oregon R*) and larvae with Hml+ blood cell (*HmlΔ-Gal4 UAS-GFP CG8501 RNAi*). Magnified images are on the right. (**G**) A visualization of the phagocytic ability of *Drosophila* hemocytes. Hemocytes (green) showed reduced phagocytotic ability against *E. coli* (magenta, top) and *S. aureus* (magenta, bottom) in *CG8501* RNAi-expressing mutants (*HLT-Gal4 UAS-GFP CG8501 RNAi*). (**H**) Quantifications of the phagocytotic abilities of hemocytes against bacteria in panel **G** (***$p < 0.0001$). Horizontal bars indicate median values.

hemocytes. Interestingly, we observed a significant reduction in the number of $Hml^+$ plasmatocytes in *CG8501* RNAi (*HmlΔ-Gal4 UAS-GFP CG8501 RNAi*) mutants (Fig 4D and 4E). However, this genotype did not alter the numbers of $Pxn^+$ plasmatocytes, $PPO1^+$ crystal cells, or total hemocytes (S8B and S8C Fig). A similar reduction was observed in whole larvae (Fig 4F), suggesting that *CG8501* is required for maintaining the number of *Hml*-expressing hemocytes. Consistently, we validated that mRNA levels of *Hml* were also decreased in hemocytes expressing *CG8501* RNAi, concomitant with reduced *CG8501* transcripts (S8D Fig). In addition to the reduction in $Hml^+$ plasmatocytes, *CG8501* RNAi also reduced the number of Nimrod C1 (NimC1)-positive plasmatocytes (Figs 4C–4E and S8E). Downregulation of CG8501 caused an overall reduction of NimC1 at the membrane; however, the NimC1 expression between two juxtaposed membrane regions remained relatively stable (Fig 4C). It is interesting to note that there was a significant increase in *NimC1* transcripts and the overall level of NimC1 protein by *CG8501* RNAi (*HmlΔ-Gal4 UAS-GFP CG8501 RNAi*), contrasting with the NimC1 expression at the hemocyte membrane (S8D and S8E Fig). This incompatibility suggests that the loss of CG8501 alters the membrane localization of NimC1 in hemocytes, which in turn induces *NimC1* transcription and accumulates NimC1 proteins in larval hemocytes.

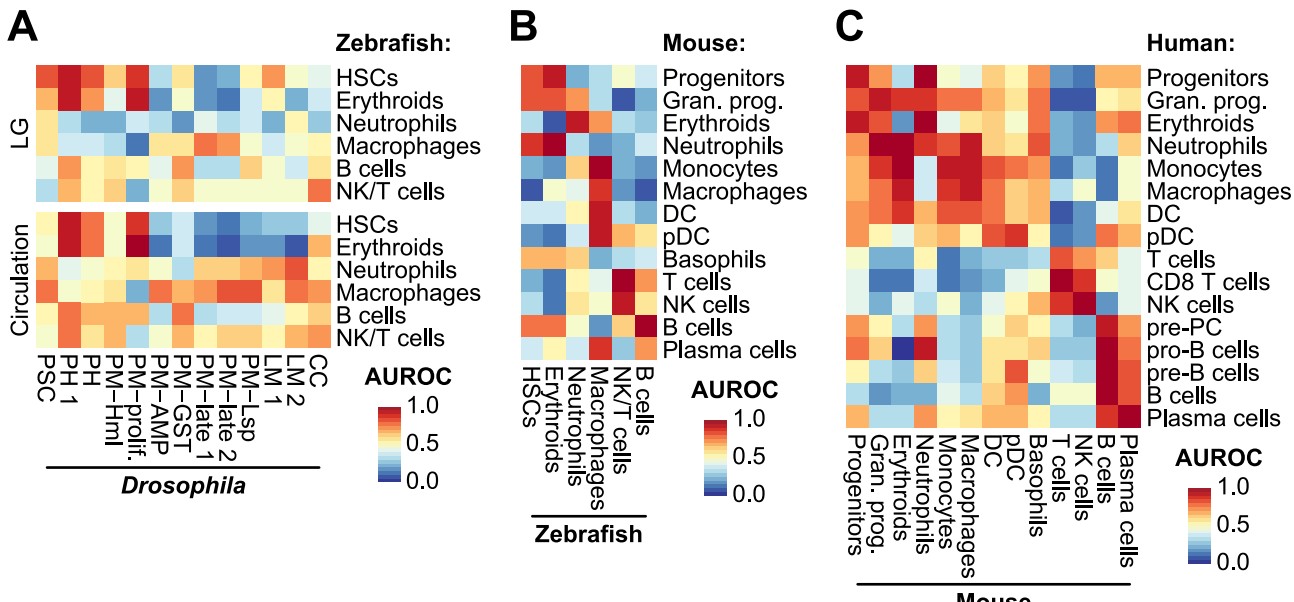

**Fig 5. Unsupervised cross-species analysis using MetaNeighbor.** MetaNeighbor AUROC values calculated using (**A**) *Drosophila* and zebrafish, (**B**) zebrafish and mouse, and (**C**) mouse and human immune cells. The MetaNeighbor analysis was performed using the pseudo-cell transformed expression data of the orthologous genes.

Future studies will elucidate the significance of the transcriptional feedback loop involving *NimC1* and the distinct regulatory role of CG8501 in NimC1 membrane localization.

*Drosophila* NimC1 is a well-known transmembrane receptor expressed in hemocytes that is critical for bacterial phagocytosis [6,34]. To validate whether *CG8501* plays a role in phagocytosis associated with NimC1 and Hml expression, we cultured wild-type or *CG8501* RNAi hemocytes with bacteria *ex vivo* (*HLT-Gal4* UAS-*CG8501 RNAi*) (Fig 4G and 4H). Hemocytes expressing *CG8501* RNAi showed significantly decreased phagocytosis activity against both Gram-positive *Staphylococcus aureus* and Gram-negative *Escherichia coli* (Fig 4G and 4H). Overall, these findings indicate that CG8501 is required for the membrane expression of the phagocytotic receptor NimC1 as well as for the expression of *Hml* in hemocytes, which are crucial for their phagocytotic function.

## Transcriptome-wide similarities between immune cells

To further compare immune cell types across species, we leveraged an unsupervised approach using MetaNeighbor [35], which was used in a recent study to compare various model species at the atlas level [36]. MetaNeighbor predicts a cell's type based on neighboring cells in the latent space and reports its confidence using the area under the receiver operating characteristic (AUROC). We used loose AUROC thresholds in this analysis because many immune cells were found in continuous rather than discrete cell clusters: 0.75 when comparing *Drosophila* to other species and 0.80 for other comparisons. The PH1 cells are a small subset of the prohemocyte population showing stem-like features. In both lymph glands and circulation, PH1 cells bore the closest resemblance to hematopoietic stem cells (HSCs) and erythroid cells from zebrafish (Fig 5A). Plasmatocytes, which constitute the most abundant type of hemocyte in *Drosophila*, have been proposed to share functional similarities with mammalian innate immune cells, including macrophages [2]. Our analysis confirmed that plasmatocytes from the 120 h AEL timepoint (PM-late2) and plasmatocytes expressing Lsp (PM-Lsp), whether in circulation or the lymph glands, showed the closest transcriptional resemblances to zebrafish macrophages. In contrast, proliferative plasmatocytes (PM-prolif) displayed a lesser degree of similarity to zebrafish macrophages but were similar to HSCs and erythroids (Fig 5A). This difference could be attributed to the proliferative characteristics shared between stem-like PH1 and PM-prolif cells, which initiates a developmental continuum of plasmatocyte differentiation in both the lymph gland and circulation [14,17]. In contrast, crystal cells and lamellocytes displayed transcriptional homologies with NK/T cells or neutrophils (Fig 5A). Moreover, AUROC thresholds for these cell types varied based on their origins, suggesting that the transcriptional characteristics of crystal cells and lamellocytes are not as distinct as those of plasmatocytes or prohemocytes.

Most immune cell types from zebrafish and mice showed many molecular similarities with the same cell types in mice and humans, respectively, indicating that molecular features of orthologous genes are well preserved between vertebrates (Fig 5B and 5C). For example, erythroids, neutrophils, and monocytes or macrophages from zebrafish and mice were matched to the same cell types from mice and humans, respectively. The NK and T cells from mice and humans formed a continuous cluster with shared transcriptomic features (Figs 3D, 3E and S5B–S5E), and the MetaNeighbor analysis also predicted similarities between these cell types (Fig 5B and 5C).

We also applied a supervised analysis by evaluating the enrichment of marker gene expression using gene set variation analysis (GSVA) [37]. First, cell type markers were studied in *Drosophila*, zebrafish, and mouse immune cell types by performing differentially expressed gene (DEG) analyses at the single cell level using MAST [38]. The GSVA performed on data

from the more complex organisms using marker genes from the simpler model organisms (S9A, S9B and S10A Figs). For example, the expression of cell type markers from *Drosophila* was investigated in zebrafish. We found that marker genes of *Drosophila* PH1 cells and multiple plasmatocyte subtypes were expressed in zebrafish HSCs and macrophages, respectively (S10A Fig). However, certain plasmatocyte subtypes, such as PM-Hml or PM-AMP, from both the circulation and lymph gland resembled macrophages, a relationship that was not detected in the MetaNeighbor analysis. We repeated the comparative analysis based on annotations from Cho et al. and found similar trends; plasmatocytes from 120 h AEL and earlier developmental time points (PM in S9C Fig), in both circulation and lymph glands, showed similarities with macrophages (S9C Fig) [12]. For other cell types, there was a strong overlap between markers in *Drosophila* PHs and B and NK/T cells from zebrafish, which could be partly explained by markers co-occurring in NK/T or B cells and vertebrate progenitor cells (S9A and S9B Fig). This relationship was weakly observed in the unsupervised analysis (Fig 5A and 5B). Other molecular homologies between cell types in vertebrates that were observed in the supervised analysis were similar to those in the unsupervised analysis. Taken together, the results from both unsupervised and supervised analyses predicted largely similar trends: immune cell types in *Drosophila* show conservation with the innate immune cells from zebrafish, including macrophages, and PH1 cells were similar to HSCs at the molecular level.

## *Drosophila* hemocytes are preserved as myeloid cells in vertebrates

To summarize similarities between immune cells, we retained only high-confidence cell type pairs (Fig 6), which were defined as cell types with average scaled MetaNeighbor AUROC values and scaled GSVA enrichment scores above a 0.80 threshold or with reciprocal best hits in the MetaNeighbor analysis (see Methods section). For example, *Drosophila* PH1 cells showed the highest conservation score with HSCs followed by erythroids in zebrafish. Likewise, PM-late1, PM-late2, and PM-Lsp cells of larvae showed greater similarities with vertebrate monocytes, illustrating features shared by *Drosophila* hemocytes and innate immune cells in more complex organisms. We additionally analyzed the 13 *Drosophila* hemocyte clusters to immune cells of mice and humans and found similarities between comparable cell types (S10B and S10C Fig). In these results, PM-late2 and PM-Lsp clusters, which included most of PM 120 h AEL, showed the highest similarities with vertebrate myeloids.

In addition to the previous datasets acquired from the *Oregon R* strain, we sequenced 2195 circulating hemocytes from the $w^{1118}$ strain using a different droplet-based scRNA-seq platform (10X Chromium 3'-seq) to test whether the results could be reproduced in a different genetic background. The population mostly consisted of plasmatocytes (*n* = 1620), while a small number of PSC cells, PHs, and CCs were also detected compared to the numbers seen in the Drop-seq datasets (S11A–S11C Fig). Although it was unlikely that we would observe lamellocytes in healthy animals, we did identify both lamellocyte subtypes (80 cells), indicating that cells may have experienced stress during sample preparation [39]. We performed marker gene enrichment and clustering-based prediction analyses with the same criteria as described above and found that late-stage plasmatocyte subtypes, such as PM-late2 or PMs (120 h AEL), and PH1 of Cho et al.'s annotation were matched to zebrafish macrophages and HSCs, respectively (S11D Fig) [12]. These results confirm that the major populations of *Drosophila* hemocytes show similarities with myeloid cells in vertebrates.

## Discussion

We performed a cross-species comparative analysis of the hematopoietic system, utilizing scRNA-seq datasets from *Drosophila* and three other vertebrate organisms. First, we carefully

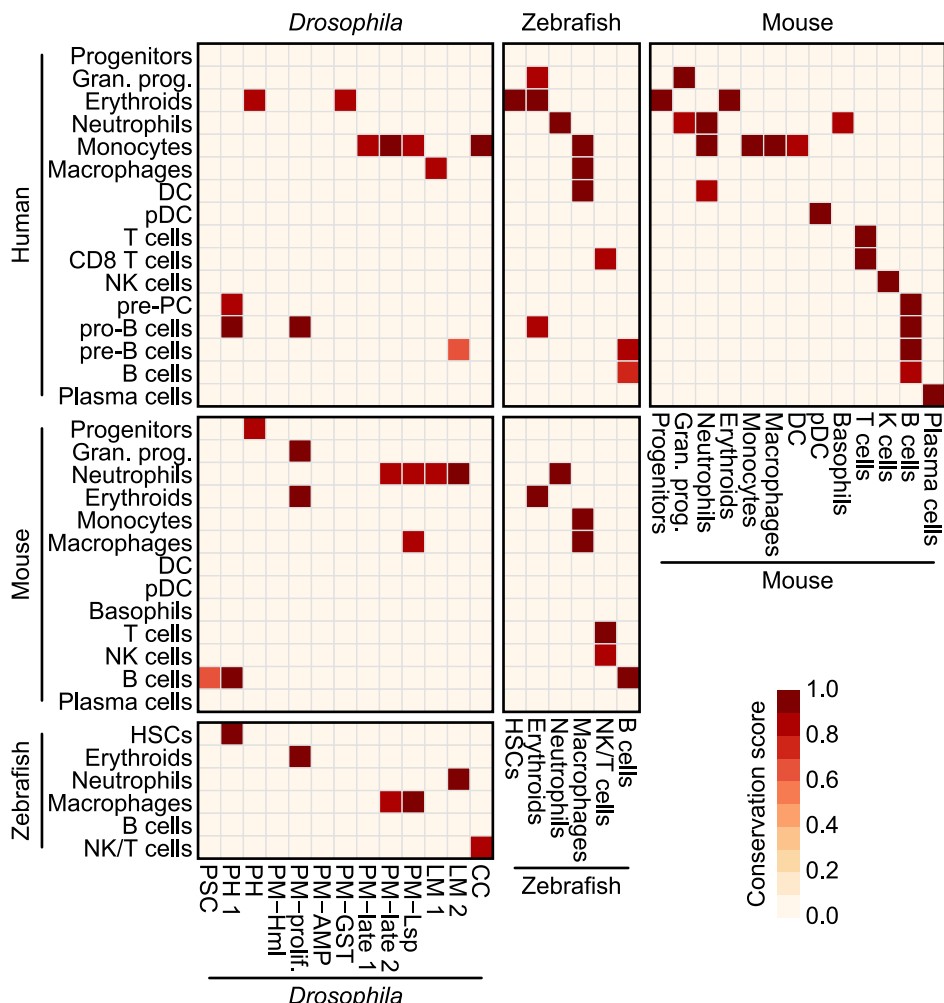

**Fig 6. Conservation map of immune cells across species.** A conservation score heatmap predicted by integrating MetaNeighbor predictions and GSVA scores. Conservation scores were calculated by averaging MetaNeighbor AUROC and scaled GSVA scores for each cell type pair. Only cell type pairs assigned with reciprocal best hits by MetaNeighbor or conservation scores above 0.8 were included.

integrated data from *Drosophila* lymph glands and circulating hemocytes collected under normal and wasp-infected conditions at specific developmental time points based on the cell types identified in the lymph gland. Next, we integrated and compared data from six publicly available scRNA-seq studies of *Drosophila* hemocytes to provide a comprehensive profile with all available information. By employing uniform parameters, we successfully merged transcriptome profiles from different datasets and classified hemocyte cells into 17 distinct clusters, including six plasmatocytes, two lamellocytes, two prohemocytes, and one PSC cell subtype. Furthermore, through a cross-species analysis, we confirmed the similarity between plasmatocytes and vertebrate myeloid cells, particularly macrophages or monocytes, as shown by the expression of several functionally conserved genes. In addition, our investigation revealed intriguing transcriptional similarities between stem cell-like PH1 cells and vertebrate HSCs and progenitor cells. Lastly, we identified the conservation of a plasmatocyte marker, *CG8501*, with CD59 in vertebrates, and demonstrated its role in phagocytosis and the development of Hml⁺ hemocytes.

After several single-cell transcriptome analyses of *Drosophila* hemocytes [12,14,17–20], additional efforts have been made to provide a comprehensive view of these previous studies [16,40]. Cattenoz and colleague performed a detailed comparison between three studies focusing on circulating hemocytes and classified them based on distinct marker genes [40]. Their approach consistently categorized lamellocytes and crystal cells, a phenomenon also observed in this study. Plasmatocytes, on the other hand, exhibited less consistent correlations across studies but could be classified into five subgroups using representative marker genes. Our systematic analysis, conducted with a unified annotation, discovered remarkable consistency among all datasets, leading us to subdivide plasmatocytes into seven groups, largely reflecting the five subgroups identified in the previous review [40]. For example, PM-prolif represents a proliferative subgroup commonly annotated in most studies, while PM-AMP designates plasmatocytes expressing antimicrobial peptides. The group PM-Lsp shares similar markers with secretory plasmatocytes, and PM-Hml indicates specific but less differentiated plasmatocytes, possibly demonstrating a higher degree of plasticity. Notably, we consistently identified hemocytes expressing the PSC marker genes across all datasets, regardless of their origins. This observation aligns well with the presence of Antp[+] hemocytes in adult [41,42] and pupal hemocyte populations [43]. These findings together suggest that Antp[+] hemocytes should be considered a valid population in larval circulation. Furthermore, our study confirms the transcriptional similarities between circulating and lymph gland hemocytes, utilizing datasets acquired from multiple analytical platforms. The cross-comparison of two lymph gland transcriptome profiles validated largely identical clustering. The collaborative efforts to establish a transcriptional classification of larval hemocytes have provided a solid foundation for the confident cross-confirmation of the datasets and subgroups of hemocyte types, along with their associated marker genes. These findings will stimulate future studies aimed at uncovering novel functions of hemocytes during animal development and homeostasis.

While *Drosophila* genes are well-conserved in vertebrates, only 24.09% of these conserved genes are expressed in vertebrate hematopoietic lineage cells. This limited conservation of immune-related genes between invertebrates and vertebrates can be attributed in part to the absence of an adaptive immune system in *Drosophila*. Our analysis demonstrated that orthologous genes from *Drosophila* and zebrafish successfully distinguish most immune cell types in each species. However, zebrafish lymphocytes appeared mixed together in *t*-SNE plots (Figs 3C and S5B). This suggests that while orthologous genes can distinguish lymphocytes from other immune cell types, there are no genes that distinctly describe the functions or features of T or B cells in *Drosophila* genomes. For example, recombination-activating genes, *RAG1* and *RAG2*, are absent from the *Drosophila* genome but present in zebrafish. Thus, it is possible that precursors of these genes might have invaded the genome as RAG transposons and became activated in early vertebrate ancestors, such as cartilaginous fish, prompting the emergence of the complex repertoire of B and T cell receptor genes through recombination [1].

*Drosophila* hemocytes have traditionally been considered myeloid-like cells. However, the transcriptional similarities between hemocytes and vertebrate myeloid cells have remained uncertain. Our study addressed this long-standing question and revealed several parallels between *Drosophila* hemocytes and vertebrate immune cells, particularly those belonging to the myeloid lineage. The functional resemblance between plasmatocytes and vertebrate monocyte and macrophage cells has been well-established, and our study confirms this similarity exists at the transcriptome level (Fig 6). Plasmatocytes under unchallenged control conditions are largely inactive, and the presence of plasmatocytes in late-stage larvae (PM-late1, 2, and PM-Lsp) may indicate a developmental activation of plasmatocytes prior to pupariation. Interestingly, it was these late-stage plasmatocytes that showed transcriptional similarities with vertebrate macrophages, while plasmatocytes from earlier stages did not exhibit any correlation.

It would be interesting to compare pupal and adult plasmatocytes with vertebrate macrophages to determine their similarity and compare it to that of larval plasmatocytes. Another unexpected observation is the significant similarities between some hemocyte types and vertebrate immune cells, including between lamellocytes and neutrophils. This discovery underscores the need for future investigations to correlate the functional and transcriptional similarities between *Drosophila* hemocytes and neutrophils. In addition, PH1 cells specifically showed similarities to HSCs in zebrafish. However, the relationship was less preserved in mouse and human cells (Figs 6 and S10C). This observation could be the result of two factors: 1) the number of genes with conserved expression patterns that allow similarities to be delineated between *Drosophila* cells and mouse or human cells is much lower than that between *Drosophila* and zebrafish cells and 2) the degree of progenitor clustering is different for mice and humans than it is for *Drosophila* PH1 cells or zebrafish HSCs. Because the progenitor clusters for mice and humans formed continuous trajectories with other differentiated cell types, HSCs, which are tightly maintained as a small subpopulation of progenitors, could not be detected in our clusters. However, based on *hth* expression in *Drosophila* PH1 cells or *meis1b* expression in zebrafish HSCs, which about 50% of cells in each cluster express, about 20–50% of the mouse and human progenitor cells could be defined as HSCs. Detailed subclustering of this population and additional scRNA-seq analysis of progenitor cells from zebrafish could pinpoint the exact cell types that are the most similar to the *Drosophila* PH1 cells.

In summary, our cross-species comparative analyses provide the first comprehensive snapshot of conservation between *Drosophila* hemocytes and vertebrate immune cells at the transcriptome level. We also updated Fly scRNA-seq Database 2.0 (http://big.hanyang.ac.kr/flyscrna), where users can freely explore our data to mine signature or conserved genes of interest. We anticipate that our research will help create a better understanding of *Drosophila* hematopoiesis and its relation to that of other hematopoietic systems.

## Methods

### Single-cell RNA sequencing of circulating *Drosophila* hemocytes

One hundred larvae were dissected for one scRNA-seq library. Larvae were vortexed 1 min prior to dissection, and 20 larvae were sacrificed in 10 μl of ice-cold Schneider's medium (Gibco, 21720024). Collected hemolymph was passed through a 40 μm cell strainer (Corning, 352340) and centrifuged at 7000 rpm at 4°C for 5 min. After removal of the supernatant, 1x filtered PBS was added. After cell preparation, scRNA-seq libraries were generated using the 10X Chromium 3' v2 kit (10X Genomics) following the manufacturer's protocol.

### Processing of *Drosophila* Drop-seq scRNA-seq data

Drop-seq UMI count matrices were downloaded from our previous study, accession # GSE141273 (https://www.ncbi.nlm.nih.gov/geo/query/acc.cgi?acc=GSE141273) [12]. The gene annotation version was updated using an FBgn to Annotation ID conversion table from FlyBase (https://flybase.org; fbgn_annotation_ID_fb_2019_03.tsv) to match gene IDs between datasets (BDGP 6.02 ➙ 6.22). Because the UMI matrices were already preprocessed (i.e., the filtration of low-quality cells based on UMI, gene count, and mitochondrial content thresholds), no further filtration was performed for lymph gland datasets. For circulation datasets, a few outlier cells were first filtered using UMI count thresholds: a UMI count > 50,000 for wild-type 96 and 120 h AEL cells, a UMI count > 70,000 for wasp-infected 96 h AEL cells, and a UMI count > 55,000 for wasp-infected 120 h AEL cells. Cells having UMI counts higher than two standard deviations from the mean were also removed to exclude possible multiplets. Low-quality cells were further removed using gene count thresholds: a gene count < 200 for

wild-type 96 and 120 h AEL cells, a gene count < 300 for wasp-infected 96 h AEL cells, and a gene count < 400 for wasp-infected 120 h AEL cells. The mitochondrial contents in circulating hemocytes were a bit higher than those in cells from lymph glands (in which a threshold of < 10% was used), so we applied 20% as a lower threshold.

Cell type annotations in circulating hemocytes from wild-type and wasp-infected larvae were transferred from those of the lymph gland datasets using "FindTransferAnchors()" and "TransferData()" functions with default parameters in the R package *Seurat* [44]. Minor cells (< 0.1% of the total population) and non-hematopoietic cells (posterior signaling center cells, dorsal vessel cells, ring gland cells, and neurons) in circulation datasets were removed. After these adjustments, 43,891 cells remained: 2210 from wild-type lymph glands, 72 h AEL; 9399 from wild-type lymph glands, 96 h AEL; 7783 from wild-type lymph glands, 120 h AEL; 10,158 from wasp-infected lymph glands, 96 h AEL; 995 from wild-type circulation, 96 h AEL; 1356 from wild-type circulation, 120 h AEL; 5674 from wasp-infected circulation, 96 h AEL; and 6376 from wasp-infected circulation, 120 h AEL. The UMI counts from all datasets were then normalized, log-transformed, and scaled, and a PCA analysis was performed to select the number of significant principal components (PCs, 50 in this analysis). A total of 8 datasets were integrated using Harmony with the default parameters [45], and *t*-SNE and UMAP plots were generated using the selected numbers of PCs.

### Integration of five public scRNA-seq datasets

We downloaded available raw scRNA-seq data and cell annotations from public repositories. The raw data of Fu et al. was provided by authors via personal communication [18]. The data was clustered and annotated based on markers reported in the original paper. The cell annotation of Girard et al. was provided by the first author via personal communication [19]. All raw *Drosophila* datasets were analyzed using the same genome version (BDGP 6.22, accession code: GCA_000001215.4) for fair comparison, except for two InDrops samples from Tattikota et al., due to technical issues in the analytic pipeline [14]. Processed count data were downloaded for these samples and the gene annotation was updated to be compatible with BDGP 6.22 by matching gene IDs.

All datasets were aligned to the *Drosophila* genome and quantified using CellRanger with the reference genome (BDGP 6.22) and matched gene annotations. The resulting UMI count matrices were analyzed using *Seurat* v4. To filter low-quality cells, library-specific thresholds for gene counts and proportions of mitochondrial (MT) genes were used: ≥ 500 genes and < 20% MT for Cattenoz et al. [17]; ≥ 500 genes and < 10% MT for Fu et al. [18]; ≥ 1500 genes and < 5% MT for Girard et al. [19]; ≥ 200 genes and < 40% (C1_Uninf, C3_Inf) or 30% MT (others) for Leitão et al. [20]; ≥ 250 genes (replicate 1) or 500 genes (replicate 2) and < 25% MT for the 10X data of Tattikota et al. [14]; and ≥ 500 genes (replicate 3) or 100 genes (replicate 4) and < 20% MT for the InDrops data of Tattikota et al. For each sequencing library, cells having UMIs higher than the mean + 2 standard deviations were removed (S3A Fig and S2 Table).

The cell annotations of Cattenoz et al., Girard et al., Leitão et al., and Tattikota et al. were assigned by matching barcode sequences, while cells that were additionally included in this study were inferred using label transfer analysis [14, 17, 19, 20]. The scRNA-seq data of Fu et al. was clustered at a resolution of 0.3 and annotated using marker genes reported in the original study [18]. For each dataset, label transfer analysis was performed to infer cell type/ state annotations between studies. A total of 128,542 cells from five public datasets and Drop-seq datasets were integrated using Harmony. Sixty-one PCs were used to cluster cells at a resolution of 0.5, identifying 17 clusters. Based on the marker gene expression and annotations

from the previous studies, six major cell types (PSC, PH1, PH, PM, LM, and CC) were identified, showing the highest diversity in plasmatocytes. Four small clusters were removed in the subsequent analyses: two clusters originating from InDrops from Tattikota et al. [14] ("Hsp" and "Unknown" in Fig 2); another cluster enriched with muscle-specific marker genes, such as *Mlc1* or *Mlc2* ("Muscle" in Fig 2); and the last cluster, originating primarily from Leitão et al. [20], was enriched with male-specific genes, such as *Mst84Da* or *S-Lap7* ("S-Lap" in Fig 2).

## Processing the *Drosophila* 10X Chromium scRNA-seq data

Three 10X Chromium scRNA-seq datasets were aligned to the *Drosophila* reference genome (BDGP 6.22) and quantified using CellRanger v3.1.0. The UMI count matrices were aggregated and possible doublets having UMI counts higher than two standard deviations from the mean and low-quality cells (mitochondrial contents > 10% or gene counts < 200) were removed using *Seurat*. Data from the remaining 2216 cells were normalized, log-transformed, and scaled and then annotated using label transfer based on lymph gland and circulating cells. As described in the subsection "Processing of *Drosophila* Drop-seq scRNA-seq data," cell type annotations were transferred from those of the lymph gland datasets using "FindTransferAnchors()" and "TransferData()" with the default parameters in *Seurat*, and cells mismatched between cell type and subcluster were removed (*n* = 21). Independent datasets were integrated using Harmony with the default parameters and 25 significant PCs were used for dimension reduction analyses and clustering.

## Curation and re-clustering of the public datasets

Zebrafish InDrop-seq processed UMI count data published by Tang *et al.* [22] were downloaded from the GEO repository (https://www.ncbi.nlm.nih.gov/geo/query/acc.cgi?acc=GSE100910), and cell types were annotated based on the original study. Five reported immune cell types (HSCs, erythroids, neutrophils, macrophages, NK/T cells, and B cells) were identified and stromal cells were excluded.

Mouse immune cell data from the MCA and Tabula Muris were downloaded from figshare (https://figshare.com/articles/dataset/HCL_DGE_Data/7235471 and https://figshare.com/articles/dataset/Robject_files_for_tissues_processed_by_Seurat/5821263, respectively) [23, 24]. Only data from peripheral blood or bone marrow cells were retrieved and re-clustered from each dataset based on the provided cell type annotations and the expression of known marker genes from the literature. Neutrophils in the MCA were removed because they were sparsely clustered. Ten cell types for 9165 MCA cells and 11 cell types each for the Tabula Muris 10X Chromium 3'-Seq and Smart-Seq2 datasets (3652 and 5037 cells, respectively) were defined with different cell type compositions. These three independent datasets were subsequently integrated using Harmony and visualized using UMAP. We additionally filtered cells that clustered with cell types that differed from those to which they were originally assigned, resulting in 13 different cell types with 15,168 cells in total (S5F Fig, 8191 MCA, 3427 Tabula Muris 10X, and 4764 Tabula Muris Smart-Seq2 cells).

Human immune cell datasets from the HCL and HCA censuses of immune cells were downloaded from the corresponding repositories (https://figshare.com/articles/dataset/HCL_DGE_Data/7235471 and https://data.humancellatlas.org/explore/projects/cc95ff89-2e68-4a08-a234-480eca21ce79, respectively) [21,25]. The 21,568 cells represented in the HCL data were re-clustered into 11 cell types based on the provided cell type annotations and known marker genes from the literature. Bone marrow cells represented in the HCA were quality-controlled based on UMI and gene count thresholds. First, outlier cells with > 80,000 UMIs or < 500 genes were filtered. Second, cells having UMI counts higher than two standard

deviations from the mean or mitochondrial contents > 10% were removed. Data from the remaining cells were normalized, scaled, and batch-corrected using the *Seurat* alignment method with the default parameters. Clustering was performed using 59 PCs and a resolution of 0.7. Thirty-two clusters were annotated based on known marker genes and compared to the HCL data. One stromal cell and two doublet clusters simultaneously expressing B cell/erythrocyte and monocyte/B cell markers were removed. The remaining 243,398 cells were categorized as one of 15 cell types. Two human scRNA-seq datasets were also integrated using Harmony. We filtered cells that clustered with cell types that differed from the original annotation labels. Nineteen platelet cells were also removed because the number of cells was insufficient to represent the cell type in the pseudo-cell transformed data. The 262,630 integrated cells (242,472 HCA and 20,158 HCL cells) were categorized as 16 different hematopoietic cell types (S5G Fig).

## Correlation analysis between mouse and human public datasets

For mouse and human immune cells, normalized expression matrices were extracted using the "GetAssayData()" function with "slot = 'data'" in *Seurat* and antilogged using "exp()". Pseudo count 1 was subsequently subtracted. Then, the single-cell expression values were averaged into pseudo-bulk values for each cell type. Cell type pseudo-bulk expression values were compared using Spearman correlation analysis and visualized using the *pheatmap* package.

## Searching for orthologous genes between species

For each of the four species, we searched lists of orthologous genes between all possible species pairs using the Drosophila RNAi Screening Center Integrative Ortholog Prediction Tool (DIOPT; http://www.flyrnai.org/diopt) [26]. When a gene in species A was matched to multiple genes in species B (one-to-many) or multiple genes in species A were matched to a single gene in species B (many-to-one), we chose the pair with 1) the highest DIOPT weighted score, 2) "best score reverse = = Yes," and 3) the highest expression level in the corresponding scRNA-seq data. We retrieved all one-to-one matched pairs.

## *t*-SNE dimensionality reduction analysis

A *t*-SNE analysis was performed on the normalized expression values using the R package *Rtsne* with "dims = 3" and "pca_scale = FALSE" or default parameters [46]. The expression matrix was extracted using "GetAssayData()" with "slot = 'data'" in *Seurat*, and only expressed orthologous genes were included. A full dataset was used for zebrafish, whereas mouse and human data were downsampled to one-tenth (3034 cells) and one-fiftieth (5253 cells), respectively. The analysis was repeated five or ten times with random seeds.

## Supervised inter-species comparisons using gene set variation analysis (GSVA)

To compare cell types using signature gene sets, a differentially expressed gene analysis was performed for each species in *Seurat* using "FindAllMarkers()" with the parameters "min. pct = 0.25," "only.pos = TRUE", and "test.use = 'MAST'." Signature genes were filtered using adjusted *P*-values ("p_val_adj < = 0.05"). Signature genes were excluded because they were identified as markers for multiple cell types due to continuously differentiating or developing cellular states in the hematopoietic organs. On average, there were 131.23 signature genes across datasets, and the number varied between different cell types, ranging from 25 to 590

genes. We performed gene set variation analysis using the *GSVA* package in R [37]. The GSVA scores were normalized to a scale of 0–1 to compare AUROC values in the following analysis.

## Unsupervised inter-species comparisons using MetaNeighbor

Unsupervised analysis was inspired by Wang *et al*. [36], and we also leveraged the MetaNeighbor approach [35]. First, normalized expression matrices were extracted using "GetAssayData ()" with "slot = 'data'" in *Seurat* and antilogged using "exp()," with pseudo count 1 subsequently subtracted. The single-cell expression data were then transformed into pseudo-cell expression data by aggregating data from 10 randomly selected cells in each cell type. By doing so, the complexity of the individual cell transcriptomes was increased while the size of the dataset was compressed to about one-tenth of its original size. The pseudo-cell expressions of species pairs were merged using orthologous genes, and MetaNeighbor analysis was performed using the "MetaNeighborUS()" function with default parameters, and variable genes were identified using "variableGenes()." Confident cell type pairs between species were selected based on AUROC values using the "topHits()" function with "threshold = 0.75" for *Drosophila*-to-zebrafish or "threshold = 0.80" for other species pairs.

The AUROC values of confident cell type pairs were then averaged with the scaled GSVA scores of the corresponding cell type pairs. Only cell type pairs reported as "Reciprocal_top_hit" in the MetaNeighbor analysis or having an average score higher than 0.80 were visualized.

## *Drosophila* genetics

These *Drosophila* stocks were used in this study: *HmlΔ-Gal4 UAS-EGFP* (S. Sinenko), *HmlΔ-Gal4 UAS-flp, Actin-FRT-Stop-FRT-GAl4, UAS-EGFP (HLT-Gal4 UAS-GFP)* (U. Banerjee), *Oregon R* (BL5), *w1118; Df(2R)BSC859/SM6a* (BL27929), *w1118; Df(2R)BSC262/CyO* (BL23297), *w1118; Df(3L)BSC393/TM6C* (BL24417), *CG8501* RNAi (NIG; 8501R-2).

## Hemocyte bleeding and staining

Bleeding followed a previous method [14]. Around 20 larvae were vortexed for one minute with glass beads (Sigma G9268) and bled on a glass slide (Immuno-Cell Int.; 61.100.17) for 40 min at 4˚C. Hemocytes were fixed with 3.7% formaldehyde for 30 min at room temperature and washed 3 times in 0.4% Triton X-100 in 1x PBS for 10 min. Hemocytes were blocked in 1% BSA/0.4% TritonX in 1x PBS for 30 min. Samples were incubated at 4˚C overnight for the primary antibody incorporation. Hemocytes were then washed 3 times in 0.4% Triton X in 1x PBS and secondary antibody treatments were performed in 1% BSA/0.4% Triton X in 1x PBS for 3 hours at room temperature. After washing 3 times with 0.4% Triton X in 1x PBS, samples were stained and mounted in Vectashield (Vector Laboratory) with DAPI. Images were captured using a Nikon C2 Si-plus confocal microscope. The antibodies CD164 (Abcam, ab238748), CD63 (Abcam, ab216130), CD59 (Invitrogen, PA5-97565), NimC1 (I.Ando), PPO1 (I.Ando), Phalloidin (Invitrogen, 22287), and Cy3- and FITC- conjugated secondary antibodies (Jackson Laboratory; 115-165-166, 711-165-152, 115-095-062, 711-095-152, 715-605-151) were used for staining at a 1:250 ratio.

## Phagocytosis assay

To check the phagocytotic ability of hemocytes, we followed a previously described phagocytosis assay method [34]. Instead of using *HmlΔ-Gal4* fly lines, we used *HLT-Gal4* fly lines for the constant Gal4 expression in hemocytes. *Escherichia coli* BioParticle (Invitrogen P35361) and

*Staphylococcus aureus* BioParticle (Invitrogen A10010) were used in separate assays. Larvae from 120 h AEL were bled and incubated in Schneider's medium containing 1ug/ml BioParticle for 30 min at room temperature. Then, hemocytes were fixed with 3.7% formaldehyde for 30 min at room temperature on glass slides (Immuno-Cell Int.; 61.100.17) and washed 3 times in 0.4% Triton X-100 in 1x PBS for 10 min. After washing, samples were mounted in Vectashield and imaged using a Nikon C2 Si-plus confocal microscope. BioParticle uptake by hemocytes was counted using the IMARIS software (Bitplane).

## RT-qPCR

At least 100 larvae were dissected to extract hemocyte mRNA, and cDNA was synthesized with a qPCR RT kit (TOYOBO). The RT-qPCR was performed by the comparative Ct method using SYBR Green Realtime PCR Master Mix (TOYOBO) and a StepOne-Plus Real-Time PCR detection thermal cycler (Applied Biosystems). Nine primer pairs were used for this analysis (F means forward, R means reverse, and all primers are written in the 5' to 3' direction):

Rp49: F- GGCCCAAGATCGTGAAGAAG, R- ATTTGTGCGACAGCTTAGCATATC.

Hml: F-GTAAGGGTCCCAACTGCGTA, R-CTGGAATGTGTGGACACCAG.

Pxn: F- ATCACGTGGATGCACAACAC, R- CGAATCGAGTGGGTGGTTAC.

CG8501-1: F: CGAGTGTGTCGATCAGGAGA, R: GCTCCCAATGCTTTCCAATA.

CG8501-2: F: GCTGACCACAATGGTGAATG, R: GACCAGGGCCAATAAGATCA.

eater-1: F: TTAATTGTGGAAGTGGCTTCTGC, R: GGTTCCTCGACTACATCCCTTG.

eater-2: F: CCTCGGACTCGTATCGGCT, R: GCAGCAATCCCTCGTTTGAAC.

NimC1-1: F: GAGACTGCCTACAGGACCGTA, R: GCAGAATCCATGTTGAGGACAC.

NimC1-2: F: TCCTCAACATGGATTCTGCTCG, R: CAAACGGGATGGCAGTCGATA.

## Western blotting

To extract proteins from *Drosophila* hemocytes, 50 larvae were bled in Schneider's medium. After bleeding, hemocytes were filtered through a 40μm cell strainer and centrifuged at 4˚C and 6000 rpm for 5 min. Cell pellets were lysed with RIPA buffer (MB-030-0050, Rockland) containing a protease inhibitor cocktail (P9599, Sigma). The protein concentrations of samples were measured using Bio-Rad Protein Assay Dye Reagent (5000006, Bio-Rad). The antibodies anti α-Tub (DSHB 12G10, 1:1000) and anti-NimC1 (I. Ando, 1:1000) were used for Western blot analysis. To detect NimC1 protein, we performed a non-reducing SDS-PAGE by excluding the reducing reagent (β-mercaptoethanol).

## Code availability

In-house R and Python codes used in this study are available on GitHub (https://github.com/sangho1130/dmel_cross_species). The detailed parameter settings and thresholds used in the analyses are described in the Methods section. All analyses were performed using Python (version 2.7.5), R (version 3.5.3), and R Studio (version 1.1.383). Detailed software versions are also described in the Methods.

## Supporting information

**S1 Fig. Integration of *Drosophila* Drop-seq datasets.** (**A**) UMAP plots of the seven major hemocyte types at three developmental timepoints (h AEL: hours after egg laying) in wild type (left) and wasp-infected (right) larvae. (**B**) UMI and (**C**) gene counts in each time point, tissue origin, and infection treatment.
(EPS)

**S2 Fig. Expression of characteristic markers.** Expression of the canonical marker genes of (**A**) PSC, (**B**) LM, (**C**) CC, and (**D**) PM, as curated by Hultmark and Andó [16]. The dot color indicates average levels of expression, and the dot size represents the percentage of cells expressing the gene in each cell type. Expression levels are shown for wild type (WT) and wasp-infected (inf) larvae.
(EPS)

**S3 Fig. Integration of public *Drosophila* scRNA-seq datasets.** (**A**) The UMI (top), gene counts (middle), and mitochondrial genes (bottom; %) in each dataset. (**B**) UMAP plots of hemocytes categorized by broad cell types (top), experimental conditions (middle), and tissue origins (bottom). (**C**) Proportions of prohemocytes (PH) in circulation (left) or lymph glands (right) in each experimental condition.
(EPS)

**S4 Fig. Comparisons of cell annotations between scRNA-seq studies.** Predictions of cell annotations using label transfer analysis. Cell annotations were predicted using five public scRNA-seq studies and compared to our cell annotations (**A**) or vice versa (**B**).
(EPS)

**S5 Fig. Re-clustering of public datasets.** UMAP plots of re-clustered cell types for (**A**) zebrafish, (**B**) mouse, and (**C**) human scRNA-seq data. Heatmaps of Spearman correlation coefficients between different datasets or platforms in (**D**) mice and (**E**) humans. Integrated scRNA-seq data for (**F**) mice and (**G**) humans.
(EPS)

**S6 Fig. Identification of cell type clusters using orthologous genes.** The *t*-SNE plots of (**A**) *Drosophila* and (**B**) zebrafish scRNA-seq data using the orthologous genes between the two species. The *t*-SNE plots of (**C**) zebrafish and (**D**) mouse scRNA-seq data using their orthologous genes. The *t*-SNE plots of (**E**) mouse and (**F**) human scRNA-seq data using their orthologous genes. The mouse and human datasets were randomly downsampled to one-fifth (3034 cells) and one-fiftieth (5253 cells), respectively.
(EPS)

**S7 Fig. Identification of cell type clusters using 4267 conserved genes.** The *t*-SNE plots of (**A**) *Drosophila*, (**B**) zebrafish, (**C**) mouse, and (**D**) human scRNA-seq data using 4267 core orthologous genes between all four species. All 3301 zebrafish cells were used. The other datasets were randomly downsampled as described previously.
(EPS)

**S8 Fig. *Drosophila CG8501*, an orthologous gene of human *CD59*.** (**A**) Expression of CD orthologues in *Drosophila* hemocytes. *Drosophila* hemocytes were marked by anti-CD59 (magenta) targeting CG8501 (*Oregon R*; left top). However, this pattern disappeared in a deficiency mutant containing *CG8501* (*w^1118^*;*Df(2R)BSC859/SM6a*, left bottom). Neither CD63 (middle) nor CD164 (right) was expressed in the wild type (*Oregon R*) or *CD63* deficiency (*w^1118^*;*Df(2R)BSC262/CyO*, middle bottom) or *CD164* deficiency (*w^1118^*;*Df(3L)BSC393/TM6C*,

right bottom) mutant hemocytes. The protein F-actin was stained by phalloidin (green). (**B**) Visualization of Hml+ or Pxn+ plasmatocytes created by hemocyte-specific knockdown of *CG8501* (*HmlΔ-Gal4 UAS-GFP CG8501 RNAi*). (**C**) Quantification of PPO1+ crystal cells, Pxn + plasmatocytes, or total DAPI+ hemocytes in wild-type (*Oregon R*) larvae and larvae carrying *CG8501* RNAi (*HmlΔ-Gal4 UAS-GFP CG8501 RNAi*). These levels are related to Fig 4D (n.s: not significant, $p > 0.01$). Horizontal bars indicate median values. (**D**) Relative mRNA expression of hemocytes in *CG8501* RNAi knockdown mutants (*HmlΔ-Gal4 UAS-GFP CG8501 RNAi*). Primers for *eater*, *NimC1*, and *CG8501* were used in two different sets. Losing CG8501 led to 1.3 times higher expression of *eater*, 1.7 times higher expression of *NimC1*, and 1.3 times higher *Pxn* expression, while *Hml* transcripts decreased by 25% in *CG8501* knockdown hemocytes compared to the expression levels in controls. The RNAi efficiency of *CG8501* RNAi used in this study was ~80%. (**E**) Western blotting analysis of NimC1 and α-tubulin using *Drosophila* hemocyte extracts. Protein-level NimC1 was increased in *CG8501* RNAi knockdown mutants (right) compared to wild-type controls (left). The relative levels of NimC1 or α-tubulin are indicated above the lanes.
(EPS)

**S9 Fig. Supervised cross-species analysis using GSVA.** The GSVA results between (**A**) zebrafish and mouse and (**B**) mouse and human immune cells. The analysis was performed using pseudo-bulk transformed expression of cell types. (**C**) Cross-species analysis based on Cho et al.'s [12] cell annotations comparing *Drosophila* and zebrafish using MetaNeighbor (top) and GSVA (bottom).
(EPS)

**S10 Fig. Cross-species analysis of *Drosophila* cell types.** (**A**) Supervised cross-species analysis comparing *Drosophila* and zebrafish using GSVA. (**B**) MetaNeighbor AUROC values calculated using *Drosophila* and mice (top) or *Drosophila* and humans (bottom). (**C**) GSVA between *Drosophila* and mice (top) or *Drosophila* and humans (bottom).
(EPS)

**S11 Fig. Validation of the *Drosophila* conservation map using a different droplet-based single-cell sequencing platform and strain.** (**A**) A *t*-SNE plot of the circulating hemocytes of *Drosophila* at 120 h AEL ($n = 2195$). Data were produced using 10X Chromium 3'-seq. The cell count of each cell type is indicated in parentheses. (**B**) The UMI (left) and gene (right) counts in three independent sequencing libraries. (**C**) The proportion (left) and count (right) for each cell type from three independent sequencing libraries (different shades of green). (**D**) A conservation map of *Drosophila* hemocytes inferred by integrating GSVA and MetaNeighbor analyses.
(EPS)

**S1 Table. Top 50 marker genes for each hemocyte cell type.**
(XLSX)

**S2 Table. Metadata of six public *Drosophila* scRNA-seq datasets.**
(XLSX)

**S3 Table. Lists of all orthologous genes.**
(XLSX)

**S4 Table. List of 4267 core orthologous genes between all four species.**
(XLSX)

**S5 Table. Lists of gene ontologies enriched in vertebrate specific orthologous genes.**
(XLSX)

**S1 Dataset. Western blot raw data.**
(ZIP)

## Acknowledgments

We thank all Bioinformatics and Genomics (BIG) lab members for their inspiring comments and discussions.

## Author Contributions

**Conceptualization:** Sang-Ho Yoon, Jin-Wu Nam.

**Data curation:** Sang-Ho Yoon.

**Formal analysis:** Sang-Ho Yoon, Hanji Kim.

**Funding acquisition:** Sang-Ho Yoon, Jin-Wu Nam.

**Investigation:** Bumsik Cho, Daewon Lee.

**Supervision:** Jiwon Shim, Jin-Wu Nam.

**Writing – original draft:** Sang-Ho Yoon, Bumsik Cho, Jiwon Shim, Jin-Wu Nam.

**Writing – review & editing:** Sang-Ho Yoon, Bumsik Cho, Daewon Lee, Jiwon Shim, Jin-Wu Nam.

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
