## [Decision Letter · Decision Letter 0]

14 Jul 2023

Dear Dr Nam,

Thank you very much for submitting your Research Article entitled 'Molecular Traces of Drosophila Hemocyte Evolution' to PLOS Genetics.

The manuscript was fully evaluated at the editorial level and by independent peer reviewers. The reviewers appreciated the attention to an important problem, but raised some substantial concerns about the current manuscript. This includes a lack of specific/detailed analyses (and representation of the data in the Figures) of genes, gene sets, and molecular pathways shared across taxa that sufficiently highlight the relevance of the study, as well as discrepancies between the analyses performed in this current study and other published reports.  Based on the reviews, we will not be able to accept this version of the manuscript, but we would be willing to review a much-revised version. We cannot, of course, promise publication at that time.

If you decide to revise the manuscript for further consideration at PLOS Genetics, please aim to resubmit within the next 60 days, unless it will take extra time to address the concerns of the reviewers, in which case we would appreciate an expected resubmission date by email to plosgenetics@plos.org.

We are sorry that we cannot be more positive about your manuscript at this stage. Please do not hesitate to contact us if you have any concerns or questions.

Yours sincerely,

Jason Karpac

Guest Editor

PLOS Genetics

Kelly Dyer

Section Editor

PLOS Genetics

Reviewer's Responses to Questions

**Comments to the Authors:**

Reviewer #1: This paper presents a novel and systematic comparison of Drosophila immune cells with vertebrate immune cells using single-cell transcriptomics. Drosophila immune cells are often compared with their vertebrate counterparts, and such a comprehensive comparison is therefore much needed. Some of the claims (Drosophila hemocytes are counterparts of primarily vertebrate innate immune cells, PH1 are counterparts of progenitors, and plasmocytes to macrophages) are well supported, but the comparison of Drosophila lamellocytes to neutrophils needs additional information (gene list with cluster information - see below) and discussion. The paper is otherwise written clearly with adequate presentation in figures, the authors have used appropriate tools (I am not familiar with GSVA and MetaNeighbor analyses to assess their use) that are available online, and the original data are deposited in appropriate databases.

The authors have produced a valuable single-cell atlas of larval hemocytes of embryonic and lymph gland origin from different time points during the 3rd larval instar and also during parasitoid wasp infestation. This comprehensive atlas thus complements previous single-cell RNAseq projects of Drosophila hemocytes. All data are available in the online Fly scRNA-seq database, which is certainly a very valuable tool for researchers interested in Drosophila immunity and hematopoiesis, but not only for them.

Major points:

1. The specific genes that are shared by Drosophila and either zebrafish, mouse or human (Fig. 2b) are not listed. Only selected genes that encode CD molecules are listed in Table S1. It would be useful to create a table with all these genes and have information on their expression in each cluster of the two species being compared or to create a searchable database with this information. For example lines 318-321: We found that marker genes of Drosophila PH 1 cells, PMs (120 h AEL), and LMs were highly expressed in zebrafish HSCs, macrophages, and neutrophils, respectively, as we observed in the MetaNeighbor analysis (Fig. 4a and Supplementary Fig. 7a).” What genes are mentioned here? Lines 342-345: “Likewise, LMs were related to neutrophils in both zebrafish and mouse, and PMs and adipohemocytes from 120 h AEL larvae showed conservation with vertebrate macrophages or monocytes, illustrating features shared by Drosophila hemocytes and innate immune cells in more complex organisms.” What are the specific features or genes common to these cell types?

2. While PH1 as counterparts of vertebrate progenitors and PM as monocytes/macrophages are more convincing and consistent with previous functional observations (PM are phagocytes, for example), LM are presented throughout the paper as counterparts of neutrophils, but this is not very convincing - they share features with monocytes/macrophages as well (Fig. 5 compared to human, Fig. S9B). As it is now stated in the paper, this could be cited in the future as “lamellocytes are counterparts of neutrophils”, which would be very simplified and the authors should ensure that this is not the case. The publication contains no in-depth discussion comparing lamellocytes and neutrophils, their functioning and roles. Lamellocytes may be very specialized cells specific to only certain species of Drosophila. Again, for future studies and interpretation, it would be helpful to have a tool to look at which genes are actually common to LM and different vertebrate cell types. The authors should look into this comparison in more detail and be very careful in formulating their conclusions.

Reviewer #2: Using their own and published single-cell transcriptomic data, the authors have made an ambitious attempt to trace the relationships between blood cell types in flies and vertebrates. Much of the data are also made easily accessible in a flyscrna database. This is very helpful. The results are not entirely clear-cut, but they should still be of interest for a broad audience. However, there are some problems with the interpretations that the authors have to address before publication.

As a starting point, the authors made an integrated clustering re-analysis of their previously published single-cell data from circulating and lymph gland cells (Tattikota et al. 2020 and Cho et al. 2020). Worryingly, the resulting clusters and subclusters are have poor match with similar studies published elsewhere. In total, at least six such studies have been published, four with circulating hemocytes (Cattenoz et al. 2020, Tattikota et al. 2020, Fu et al. 2020 and Leitão et al. 2020) and two with lymph glands (Cho et al. 2020 and Girard et al. 2021). These should all be properly referred to, and the discrepancies must be discussed. The analysis described in this manuscript corresponds well with one of their own studies (Cho et al. 2020), but not with the other one (Tattikota et al. 2020). The other four studies are not even mentioned in this manuscript.

For this discussion, it may be constructive to distinguish between cell types and cell states. Cell types are more or less stably differentiated lines of cells. Lamellocytes and crystal cells are such classically defined hemocyte types, and they are are nicely supported by all six transcriptomic studies. However, the remaining perhaps 50-95% of the cells are split into various clusters and subclusters, most of them corresponding to the plasmatocyte cell type, but perhaps transiently involved in particular activities, or states, such as mitosis or antimicrobial and stress responses. Specifically, the GST cluster may correspond to cells in a state of stress, and its markers overlap partially with clusters described in Tattikota et al. 2020, and perhaps to a limited extent in other studies as well.

The PH (prohemocyte) cluster is a special case. It includes a substantial fraction of all non-lamellocyte and non-crystal cells. Surprisingly, four of the "top 5 cell type markers" for the PH cluster (Fig. 1 d) are antimicrobial peptides, otherwise characterizing minor "AMP" subclusters in the other studies. The fifth marker, CG13160, was only detected in the lymph gland, according to the data in the flyscrna database. By exclusion, the majority of cells in the PH cluster must classically be defined as plasmatocytes, since most or all of the circulating non-lamellocyte and non-crystal cells are known to express classical markers of differentiated plasmatocytes (NimC1, hml...). If true prohemocytes (i.e. undifferentiated hemocyte precursors) exist in circulation, they must be few. This problem must be properly discussed, and the "prohemocyte" terminology may be misleading.

The follow up on the CG8501 marker is very interesting. Why is this important marker not displayed in Fig. 1d, and why does the text describe it as specific for PM (120) cells? According to the flyscrna database it is a good marker for PM cells in general. On line 275, it is stated that "knock-down of CG8501 did not change the mRNA expression of NimC1 (Supplementary Fig. 6d)", but the figure shows what looks like a highly significantly INCREASED expression of NimC1. Any comment?

The PH 1 subcluster is a very interesting, case. Unlike the main PH cluster, the PH 1 cells may well correspond to a true class of prohemocytes; it is a small class, and it has a convincing overlap with vertebrate hematopoietic precursors. The presence of this subcluster among the circulating hemocytes suggests that some prohemocytes may after all be present in that population. However, cells similar to PH 1 were never detected in the other published single-cell studies, not even in the paper by Tattikota et al. 2020. How come?

By the way, are the PH 1 cells included among the cells of the PH cluster, or should I understand these categories as mutually exclusive?

Another discrepancy involves the hemocytes related to the cells of the posterior signalling center (PSC). Most of the previous studies found a well-defined class of such cells among the circulating cells, corresponding to a separate hemocyte type, dubbed "primocytes" by Fu et al. 2020. This class was also standing out in the data of Tattikota et al. 2020 (the "PM11" subcluster), but although the same data are included in the present manuscript, the "PSC" cluster is not represented among the circulating hemocytes. Why not?

The central part of this study involves the comparison between blood cell types (and states) in Drosophila and vertebrates. The most consistent relationship shown here is between Drosophila PH 1 cells and various vertebrate hematopoietic stem cells or precursors. This makes much sense, and I look much forward to future characterization (and confirmation) of the PH 1 class.

Other correlations between the transcriptomes of Drosophila and vertebrate cell types are more uncertain. In general, they tend to link Drosophila hemocyte types to different vertebrate myeloid cells, but in some cases also to lymphoid cells. These correlations should be taken with several grains of salt. Lamellocytes are for instance strongly linked to Zebrafish and mouse neutrophils but to human monocytes Fig. 8). It should be noted that lamellocytes have only been found in a few Drosophila species, all closely related to D. melanogaster. In other Drosophila species they are replaced by other effector cell types, like the giant cells in D. ananassae. The transcriptomic profile of the latter cells is not very similar to that of D. melanogaster (Cinege et al. 2022). Similarly, the suggested relationship between crystal cells and Zebrafish NK/T cells or mouse "pDCs" (=plasmacytoid dendritic cells?) seems unlikely (Fig. 8). A relationship between plasmatocytes and mouse monocytes, or plasmatocytes (120 h) with macrophages or monocytes (Fig. 8) seems more likely, by the criterion of making sense. The value of this study is to point to similarities like these, but it should be pointed out that they do not necessarily imply homology (=common origin), rather than similar function. It could be speculated that ancestral blood cells had a phagocytic function, and that a phagocytic machinery has been retained in different more specialized blood cell types as well as in various "non-professional" phagocytes.

Regarding the comparisons between Drosophila and vertebrate blood cells I don't understand why the Drosophila transcriptomes were directly compared only with zebrafish. Mouse and human data were only secondarily compared with the zebrafish (Fig. 4 and Suppl. Figure 7). Direct comparisons between Drosophila and mouse or Drosophila and human were only shown in Fig. 8, although the latter was supposedly based on the comparisons in Fig. 4 and Suppl. Figure 7.

In conclusion, this is an important piece of work, trying for the first time to use transcriptomic data to identify relationships between blood cell types in insects and vertebrates. Novel findings include the possible existence of a prohemocyte class (PH 1 but, in my opinion, not PH in general) and the possible role of the CG8501 protein, but the uncertainties are not sufficiently emphasised, and the discrepancies between this study and those done elsewhere must be mentioned and discussed.

Minor points:

As far as possible, abbreviations should always be avoided. They tend to make reading unnecessarily difficult for anyone outside the particular narrow field. Newcomers quickly loose track, and the space you save is insignificant. Specifically, when cell types are discussed, their full names (plasmatocytes, lamellocytes etc.) should be fully spelled out. However, terms like PL, LM etc. are acceptable as designations of transcriptomic clusters (which are not necessarily synonymous with the established cell types). Abbreviations like LG for lymph gland are completely unnecessary in the main text.

The word infest is used for animals and pests that invade an area or space, like in a house infested with rats. For parasites, viruses and bacteria that affect an organism, the word infect is better. What is "steady state" (Fig. 1b). Does it mean uninfected?

The resolution is too low in some figures. For instance, in Fig. 1a it is not possible to see the dots corresponding to some of the cell types. Other figures have the same problem. In Fig. 2f and g, I am unable to read the text.

Reviewer #3: In the current study by Yoon et al., entitled “Molecular Traces of Drosophila Hemocyte Evolution”, the authors have attempted a comprehensive cross species analysis between immune cells of Drosophila and vertebrate immune cells by employing the use of available single cell RNA seq data sets. As the authors compared the transcriptome of fly, fish, mouse and human immune cells, the data presented reveals common and distinguishing attributes of the respective Drosophila immune cells. Overall, through this approach the findings allude to:

1. homology of the fly immune cells to innate immune cells of vertebrates.

2. The data compares specific PH1 subset of Drosophila immune cells and reveals that this subset of prohemocyte was closest to hematopoietic progenitors and erythroid population.

3. the majority of Drosophila immune cells, which are plasmatocytes, are akin to macrophages and interestingly, the lamellocytes bear homology to neutrophils.

4. The authors also validate/annotate CG8501, which was found to be homologous to human CD59, to be important for phagocytic activity by regulating Hml and NimC1 in Drosophila.

Overall, the findings of the manuscript reveal a trend observed in immune cells of Drosophila. The large similarity of Drosophila immune cells with cells of zebrafish at a transcriptomic level is indeed intriguing. While I find the manuscript of substantial interest, but I am afraid the current draft and the manner in which it is drafted, do not deliver the information and the relevance of the analysis. My main concern is that the title of the manuscript, which is very broad and an ambitious one, but the contents of the manuscript in its current state fall short in delivering the same. The description of the data in the results section is very minimalistic when compared along side the figures, which are very elaborate. The figures are out of proportion with respect to the results section. The discussion as well, is very loose and does not really make a case for why this study is relevant for the field.

The data presented in the current state only proves the homology of fly immune cells to vertebrate myeloid lineage. While I agree this confirmation is good, but any point beyond this already established knowledge, any new additional understanding that would prove the value of Drosophila immune cells as a powerful system and relevant towards understanding vertebrate myeloid physiology is not presented or discussed sufficiently. The draft falls short in presenting the data to highlight the value of their analysis. I feel that an analysis of such a kind should enable the field with a much deeper understanding of the Drosophila immune system and empower it further to be used as a tool to address questions relevant to myeloid physiology. The finding and representation of a handful of genes with only, CG8501 and its homology with vertebrate immune cells does not sufficiently prove “Molecular Traces of Drosophila Hemocyte Evolution”. I am therefore afraid the draft in its current state does not deliver this message.

I strongly urge the authors to re-write the manuscript to elaborate and provide more detail on the data and better discussion of genes or classes that would provide newer substance and information, which is beyond proving our current knowledge. The values of cross comparing 4 model systems with details on the obtained information with a well-bodied discussion that would further empower Drosophila as a key model system to uncover myeloid physiology and function is what I strongly recommend.

There are also a few minor comments for the author to be addressed:

1) Fig S1a: In the methods it is mentioned that for the ssRNA seq 100 larvae were taken, if that is correct then circulating hemocytes in steady state in all the developmental conditions is underrepresented, which may incorporate biases in data interpretation.

2) In the manuscript author claims that CG8051RNAi do not impact the total hemocytes but significantly impact the major hemocyte population. This is not supported by any compensation of other hemocyte population.

3) FigS6b: the graph seems to be out of place as in text author is addressing Plasmatocytes while quoting this graph, which to my understanding is representing crystal cell population.

4) FigS6c: In 2nd image of the figS6C one of the larvae do not show any reduction of Hml–UASGFP positive cells as claimed in the text, author need to change the image.

5) Fig 3: Quantification for the NimC1 positive cells is missing alongside Hml positive cells, as it is one of the important finding highlighted by author, under CG8051RNAi condition.

6) Fig 3c & d: Author highlights that CG8051RNAi reduces the NimC1 protein levels (through anti-NimC1 antibody) but figS6d shows high mRNA levels. With the understanding that mRNA levels need not always be correct proxy for protein levels. This point is raised/important because author has used the same data set to support the low Hml protein levels as mRNA levels of Hml are also low. But for the NimC1 the results are contrary.

7) even though we can clearly see that there is a significant increase in NimC1 mRNA levels in FigS6d, author is claiming no change and on this basis they are claiming CG8051 is important for stabilizing NimC1, therefore it needs more explanation.

8) Hml is a common marker for all three blood cell type, author do address the impact on Plasmatocytes population with the help of NimC1 but what are the consequences of CG8051RNAi on crystal cell and lamellocytes is worth understanding.

9) In material and methods infection strategy is missing.

10) In Drosophila hemocytes crystal cells are often compared functionally with platelets but this cross species analysis did not address this point, any comments on this aspect?

**Have all data underlying the figures and results presented in the manuscript been provided?**

Reviewer #1: **No: **The full list of genes that are shared by Drosophila and either zebrafish, mouse or human (Fig. 2b) and the information about their expression in clusters are not provided.

Reviewer #2: Yes

Reviewer #3: Yes

PLOS authors have the option to publish the peer review history of their article (what does this mean?). If published, this will include your full peer review and any attached files.

Reviewer #1: No

Reviewer #2: **Yes: **Dan Hultmark

Reviewer #3: No

---

## [Decision Letter · Decision Letter 1]

21 Nov 2023

Dear Dr Nam,

We are pleased to inform you that your manuscript entitled "Molecular Traces of Drosophila Hemocytes Reveal Transcriptomic Conservation with Vertebrate Myeloid Cells" has been editorially accepted for publication in PLOS Genetics. Congratulations!

Yours sincerely,

Jason Karpac

Guest Editor

PLOS Genetics

Kelly Dyer

Section Editor

PLOS Genetics

Comments from the reviewers (if applicable):

Reviewer's Responses to Questions

**Comments to the Authors:**

Reviewer #1: The authors in the revised version have addressed my previous comments - they have supplemented the searchable database in a truly substantial way, which represented a great deal of additional work, and I really appreciate that. I believe this will make the database a very useful tool.

I understand the authors' comment that "it is not feasible to list a specific set of genes from MetaNeighbor analyses". The updated tables in the database, accessed via Figshare, do eventually make it possible to find the sets of genes that are shared between clusters of cells between the species being compared. There is a bit of a circuitous route to this information, but the important thing is that the data is available.

The authors also addressed my comment (and that of other opponents) and softened the claim about the similarity of lamellocytes to neutrophils.

It's a bit sad that the discussion of the comparison between Drosophila and vertebrate immune cells has remained very limited and the authors go into virtually no specific details - I think this will sadden many readers because that's what they will be often looking for in this study. On the other hand, it's understandable, there are just too many possible specific comparisons of genes or biological processes. The study thus provides, above all, a very rich tool for further research on specific questions, which is what the authors state as their main goal.

Reviewer #2: I am impressed by the efforts made by the authors, now fully rectifying my criticism that full acknowledgement had not been given to other published single-cell transcriptomic studies of Drosophila hemocytes, and that the discrepancies between these studies have to be sorted out. In the revised manuscript, the authors have in fact integrated available data into a single comprehensive clustering analysis, and critically analysed the differences between the studies. This basically takes care of this main criticism, and the minor points are also dealt with. Therefore, I can enthusiastically endorse this new version of the manuscript.

Among the possible reasons mentioned for the discrepancies between the previously published studies, I personally believe that exact choice of parameters for the clustering analyses has been of paramount importance. These parameters may have been optimised to identify interesting subclusters within the heterogeneous plasmatocyte class, and potentially to identify new hidden cell types. This has indeed helped to illustrate the plasticity of these cells, and the different effector genes that are activated depending on the tasks that particular groups of plasmatocytes are executing. At the same time, we have missed the chance to identify general markers for plasmatocytes, including the transcription factors and signalling molecules that may be important to determine the plasmatocyte cell fate, like pebbled, Notch, klumpfuss and lozenge for crystal cells, and Antennapedia and knot for the PSC-like primocytes. With the integrated database created here, it should be possible to make such an analysis, and to identify a solid set of marker genes for the plasmatocytes. It may be beyond the scope of the present manuscript to ask for this analysis here, but I strongly urge the authors to do it at some point.

Reviewer #3: The revised manuscript has addressed all concerns raised. After the incorporation of new comparisons from various single-cell articles from the Drosophila lymph gland and circulation, the current version of the manuscript is much more intense with the discussion on the cell types across organisms.

A final suggestion that may help further even more the overall understanding of this article, is, I would suggest incorporating a model summarising their findings and conclusions.

Overall, this is a fabulous and a comprehensive manuscript, and definitely an excellent resource to the blood community.

**Have all data underlying the figures and results presented in the manuscript been provided?**

Reviewer #1: Yes

Reviewer #2: Yes

Reviewer #3: Yes

PLOS authors have the option to publish the peer review history of their article (what does this mean?). If published, this will include your full peer review and any attached files.

Reviewer #1: No

Reviewer #2: **Yes: **Dan Hultmark

Reviewer #3: No

**Data Deposition**

http://datadryad.org/submit?journalID=pgenetics&manu=PGENETICS-D-23-00549R1

**Press Queries**

---

## [Editor Report · Acceptance letter]

30 Nov 2023

PGENETICS-D-23-00549R1 

Molecular Traces of *Drosophila* Hemocytes Reveal Transcriptomic Conservation with Vertebrate Myeloid Cells 

Dear Dr Nam, 

We are pleased to inform you that your manuscript entitled "Molecular Traces of *Drosophila* Hemocytes Reveal Transcriptomic Conservation with Vertebrate Myeloid Cells" has been formally accepted for publication in PLOS Genetics! Your manuscript is now with our production department and you will be notified of the publication date in due course.

With kind regards,

Bernadett Koltai

PLOS Genetics

On behalf of:
